# Endothelial TRIM47 regulates blood-brain barrier integrity and cognition via the KEAP1/NRF2 signalling pathway in mice
Valentin Delobel [1,4], Camille Grenier[1,4], Romain Boulestreau[1], Sébastien Rubin [1], Juliette Vaurs[1], Béatrice Jaspard-Vinassa[1], Elina Casas[1], Muriel Busson[1], Cloé Combrouze[1], Carole Proust[2], Ilana Caro [2], Jean-Luc Morel [3], Bruno Bontempi [3], Aniket Mishra [2], Stéphanie Debette [2], Cécile Duplàa [1], Thierry Couffinhal [1,5] & Claire Peghaire [1,5] ✉

Cerebral small vessel disease (cSVD) is a leading cause of stroke, cognitive decline and dementia, for which no specific mechanism-based treatments are currently available. Previous genomic studies identified associations of common variants at chr17q25 with cSVD features, with converging evidence for a causal involvement of *TRIM47*, an ubiquitin ligase enriched in brain endothelial cells (ECs). In the present study, we devised a multilayered experimental plan to decipher the biological mechanisms underlying TRIM47's role in cSVD pathophysiology. *Trim47*-deficient mice, which model the human genetic anomaly, exhibit major cognitive impairments, increased blood-brain barrier (BBB) permeability, and astrogliosis, without neuroinflammation. Inducible deletion of *Trim47* in ECs recapitulates these phenotypes highlighting the contribution of endothelial TRIM47 in maintaining brain homeostasis. In vitro and in vivo data, demonstrate that TRIM47 regulates the resilience of brain ECs to oxidative stress by binding to KEAP1, stabilizing NRF2 protein levels and promoting the NRF2 pathway. Treatment with the NRF2 activator tert-butylhydroquinone prevented BBB and cognitive impairment in *Trim47*-mutant mice. By leveraging unique human proteomic data, we propose that modulation of the TRIM47/NRF2 pathway could predict an increased susceptibility to cSVD, suggesting that targeting this pathway may offer a promising therapeutic approach for vascular cognitive impairment and dementia.

Cerebral small vessel disease (cSVD) encompasses a range of pathological processes that damage arterioles, capillaries, or venules supplying brain tissue. It is a leading cause of stroke (both ischemic and hemorrhagic) and the major pathological substrate underlying the vascular contribution to cognitive decline and dementia, including Alzheimer's type dementia[1–4]. cSVD is highly prevalent in the general population with increasing age and is often covert, i.e. detectable on brain magnetic resonance imaging (MRI) before clinical symptoms manifest[5,6], thus representing a major target for prevention[7]. Key MRI-features of cSVD include white matter hyperintensities (WMH), microbleeds, lacunes and perivascular spaces[8], reflecting consequences of cSVD on the brain parenchyma.

Increasing age and high blood pressure are the strongest known risk factors for cSVD. However, while lowering blood pressure, and more

broadly optimal management of vascular risk factors, is an important approach to prevent and slow down the progression of cSVD, it is insufficient[9,10], with vascular risk factors explaining only a small proportion of WMH variability in older age[11]. Specific treatments for cSVD are currently lacking, partly because the molecular pathways driving its etiology remain poorly understood. Thus, understanding the precise molecular and cellular mechanisms underlying cSVD and its consequences on cognitive performance is an absolute prerequisite for identifying novel therapeutic targets.

Genomics is a powerful tool to identify disease pathways and has been repeatedly shown to provide a strong foundation for mechanistic studies and target discovery[12,13]. It has been estimated that genetic support for new drug efficacy more than doubles success rates in clinical trials[13–16]. Large

[1]Univ. Bordeaux, INSERM, Biologie des maladies cardiovasculaires, U1034, F-33600 Pessac, France. [2]Univ. Bordeaux, INSERM, Bordeaux Population Health, U1219, F-33000 Bordeaux, France. [3]Univ. Bordeaux, CNRS, INCIA, UMR5287, F-33000 Bordeaux, France. [4]These authors contributed equally: Valentin Delobel, Camille Grenier. [5]These authors jointly supervised this work: Thierry Couffinhal, Claire Peghaire. ✉e-mail: claire.peghaire@u-bordeaux.fr

collaborative genome-wide and whole-exome association studies on population-based cohort studies have identified numerous genetic risk variants associated with MRI markers of cSVD[3,17–20]. The most significant association with WMH volume and a composite extreme-cSVD phenotype was described at chr17q25[3,21]. Within this gene-rich locus, summary-based Mendelian randomization and profiling of human loss-of-function allele carriers in the UK Biobank pointed to *TRIM47* as the most plausible causal gene with evidence for human loss-of-function mechanisms driving the association with cSVD severity[21]. Moreover, transcriptomic data showed an enrichment of *TRIM47* expression in brain blood vessels and endothelial cells (ECs) (https://markfsabbagh.shinyapps.io/vectrdb) and preliminary functional evaluation with siRNA targeting *TRIM47* showed increased endothelial permeability in vitro[21]. In the present study, we devised a multilayered experimental plan to decipher the biological mechanisms underlying the association of TRIM47 with cSVD.

TRIM47 belongs to a subfamily of E3 ubiquitin ligases involved in many cellular and physiological processes, including cell proliferation, apoptosis, innate immunity, and autophagy. The subfamily of E3 ubiquitin ligases regulates protein degradation, cell trafficking and is also involved in DNA repair[22,23]. TRIM47 was first identified as being overexpressed in astrocytoma[24] and its role in cancers has been studied extensively. As an oncogene, TRIM47 promotes tumorigenesis, and is overexpressed in multiple cancers (breast[25], ovarian[26], colorectal[27], glioma[28,29], renal[30], gastric cancers[31]). It is also used as a prognostic biomarker for several cancers[31–34]. Previous work also demonstrated that TRIM47 regulates cellular functions associated with angiogenesis in vitro[28], with its expression being upregulated under various pathological conditions, including inflammation, hypoxia, and oxidative stress in human umbilical vein endothelial cells (HUVEC)[35]. Recently, two studies provided additional evidence for a possible role of TRIM47 in vascular biology and more specifically its deleterious effect in inflammatory and pathological contexts[35,36].

In the present study, we demonstrate a major contribution of TRIM47 to brain endothelial cell homeostasis, blood–brain barrier integrity, and cognitive function, through regulation of the antioxidant NRF2 pathway. Our data highlight the therapeutic potential of targeting the TRIM47/NRF2 protective axis in patients with cSVD and vascular cognitive impairment and dementia (VCID).

## Results

### TRIM47 is an activator of the NRF2 pathway in human brain endothelial cells

Our group has recently reported that TRIM47 is particularly enriched in brain vessels and plays a crucial role in regulating brain EC functions in vitro[21]. To identify the specific pathways regulated by TRIM47 in brain ECs, we performed bulk RNA-sequencing on human brain microvascular endothelial cells (HBMEC) treated with either control siRNA (siCtl) or *TRIM47*-targeting siRNA (si*TRIM47*) for 72 h. Transcriptomic data analysis revealed that 208 genes were downregulated while 214 genes were upregulated following *TRIM47* depletion (Fig. 1a). Pathway enrichment analysis revealed that the genes regulated by TRIM47, particularly those downregulated in *TRIM47*-deficient cells, were associated with the nuclear erythroid 2-related factor 2 (NRF2) pathway, oxidative stress (OS) and terms related to NRF2 (Nuclear receptors metapathway, NRF2ARE regulation, NRF2 transcriptional activation, NRF2 survival signaling) (Fig. 1b). Interestingly, *TRIM47* depletion did not affect the mRNA levels of the transcription factor *NFE2L2* (the gene encoding NRF2) and Kelch-like ECH-associated protein 1 (*KEAP1*), a regulator of NRF2, but it significantly downregulated key NRF2 target genes including *HMOX1*, *NQO1* and *GCLM* (Fig. 1c). Further time course experiments showed that inhibition of TRIM47 expression in HBMEC by siRNA treatment reduced *NQO1* and *HMOX1* (HO1) mRNA levels over 24, 48 and 72 h (Supplementary Fig. 1a–c) with a corresponding decrease in HO1 protein levels (Fig. 1d). Rescue experiments through TRIM47 overexpression restored HMOX1 and NQO1 expression in siTRIM47-treated cells, confirming TRIM47's key role in their regulation (Supplementary Fig. 1d).

Additionally, *TRIM47* expression was upregulated in HBMEC but *NFE2L2* mRNA levels were not affected (Supplementary Fig. 1e, f) under OS conditions induced by $H_2O_2$, corroborating previous observations in HUVEC[35]. Crucially, TRIM47 was required to fully induce *HMOX1* and *NQO1* mRNA levels in both basal and OS conditions in HBMEC, mirroring the effects of *NRF2* siRNA (Fig. 1e, f). In line with these data, $H_2O_2$ treatment activated the HO1 promoter (HO1-4.5 kb); *TRIM47* depletion caused a significant decrease in HO1 promoter activity in both baseline and OS conditions (Fig. 1g). Together, these data showed that TRIM47 is a key activator of the NRF2 antioxidant pathway in human brain ECs.

### TRIM47 regulates NRF2 stability via its interaction with KEAP1 in endothelial cells

To uncover the precise mechanism by which TRIM47 regulates the NRF2 pathway, we investigated the potential interaction between TRIM47 and the transcription factor NRF2 in brain ECs using in vitro models. *TRIM47* overexpression (Supplementary Fig. 2a) led to a modest increase in *NFE2L2* expression (Supplementary Fig. 2b) and to a significant increase in the expression of NRF2 target genes *NQO1* and *HMOX1* (Fig. 2a, b). This induction was abolished by *NRF2* siRNA treatment, showing that TRIM47's effect on these genes is dependent on NRF2 (Fig. 2a, b).

Given that NRF2 is the master regulator of antioxidant defense gene expression and directly drives the expression of HO1 (the best described NRF2 target and a direct NRF2 binding partner) and NQO1 by binding to antioxidant response elements (ARE), on their promoters (reviewed in ref. 37), we sought to determine whether TRIM47 requires these NRF2 DNA binding sites to increase HO1 expression. A luciferase assay in HeLa cells revealed that *TRIM47* overexpression activates the HO1 promoter (Fig. 2c). Mutations in the two ARE sites on the HO1 promoter (HO1-4.5 kb 2 Mut, double mutant construct)[38,39] blocked HO1 promoter transactivation, underscoring the necessity of NRF2 binding to ARE sites for TRIM47-mediated HO1 induction (Fig. 2c). Similar effects were also observed with a triple mutant construct with additional mutation in the E box of the HO1 promoter (HO1-4.5 kb 3 Mut) which fully blocked HO1 promoter transactivation induced by TRIM47 (Fig. 2c).

The activity of NRF2, both under basal conditions and in response to stress, is tightly controlled. NRF2 protein levels are finely regulated by KEAP1, a master negative regulator that binds to NRF2, leading to its ubiquitination and subsequent degradation by the proteasome (reviewed in ref. 37). To investigate whether TRIM47 modulates the NRF2 pathway through the formation of a regulatory complex, we performed co-immunoprecipitation (Co-IP) assays. Our data revealed that TRIM47 interacts with KEAP1 in HeLa cells overexpressing both 1) TRIM47-myc and KEAP1-Flag (Fig. 2d) or 2) only TRIM47-myc (semi-endogenous Co-IP) (Supplementary Fig. 3a).

Given that TRIM47 is an E3 ubiquitin ligase, we initially hypothesized that it might directly promote KEAP1 ubiquitination. However, TRIM47-myc overexpression did not increase KEAP1 ubiquitination in semi-endogenous Co-IP assays (Supplementary Fig. 3a). Ubiquitination of KEAP1 by cullin–RING ligases was also excluded, as MLN4924 treatment failed to alter KEAP1 ubiquitination levels (Supplementary Fig. 3b). Together, these findings indicate that KEAP1 ubiquitination is unlikely to mediate TRIM47-dependent regulation of the KEAP1–NRF2 pathway. Importantly, although TRIM47 depletion in HBMEC did not affect KEAP1 protein levels, Western blot analysis, with and without a proteasome inhibitor treatment (MG132), showed a significant decrease in NRF2 protein (Fig. 2e). This suggests that TRIM47 may prevent NRF2 protein degradation by binding to KEAP1. ECs were then treated with an NRF2 pathway activator, namely the tert-butylhydroquinone (tBHQ), which promotes NRF2 protein stability by destabilizing the KEAP1-NRF2 complex[40]. This NRF2 pathway activator did not affect the mRNA levels of *TRIM47* (Supplementary Fig. 2d) and *NFE2L2* (Supplementary Fig. 2e). However, importantly, tBHQ was able to partly rescue the effect of *TRIM47* siRNA on *NQO1* (Fig. 2f) and *HMOX1* (Fig. 2g) expression,

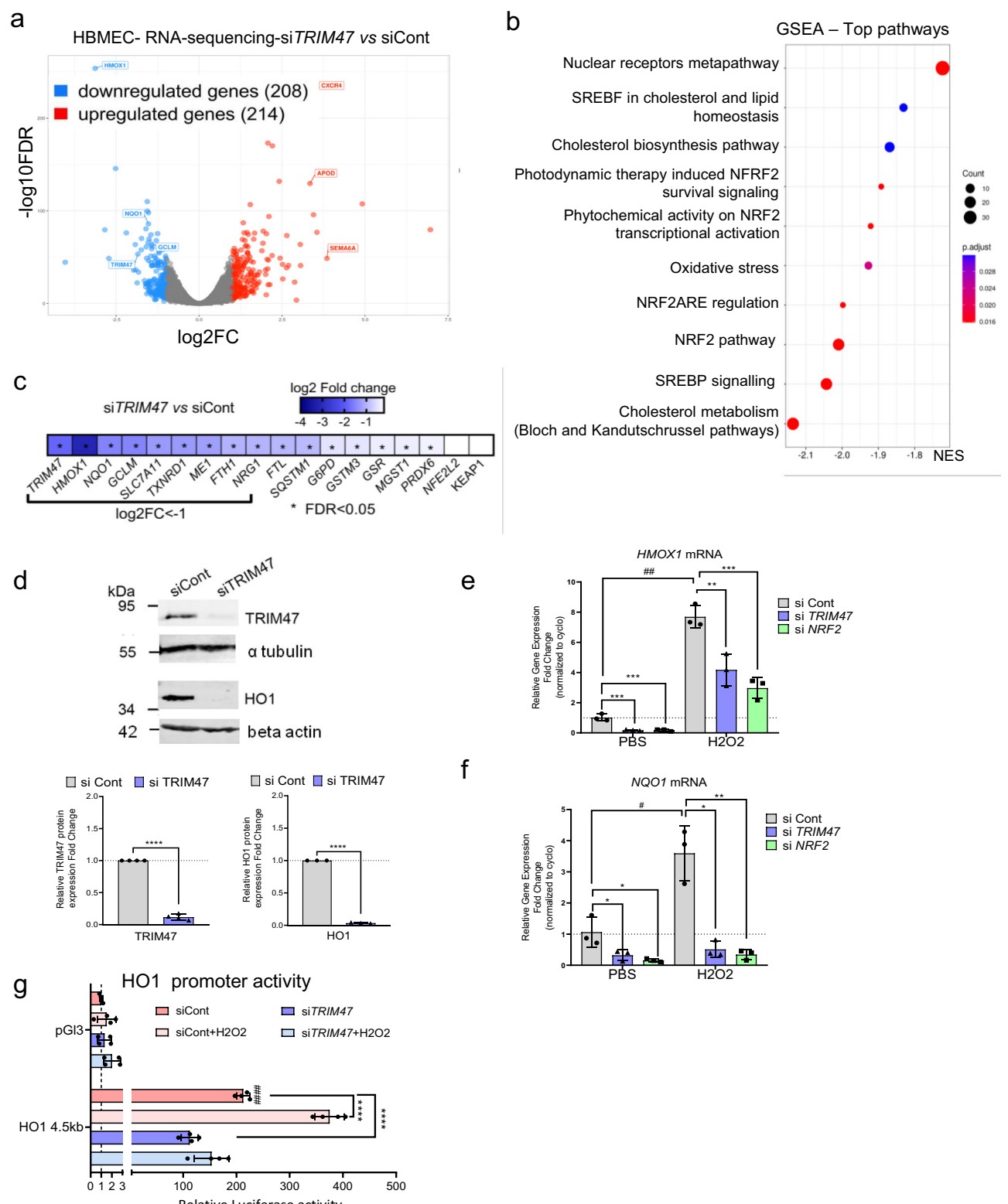

indicating that TRIM47 acts upstream of NRF2. As expected, tBHQ could not rescue the effects of *NRF2* siRNA on these target genes (Fig. 2f, g). These findings highlight TRIM47 as a key modulator of NRF2 stability through its interaction with KEAP1.

In summary, these data indicate that under basal conditions, when TRIM47 is expressed in ECs, it binds to KEAP1 in the cytoplasm, thereby reducing NRF2 protein degradation and enhancing its stability. NRF2 can

then accumulate in the cytoplasm and translocate to the nucleus and binds to ARE sites to induce the transcription of its antioxidant target genes. Conversely, when TRIM47 is repressed, KEAP1 is free to bind to NRF2, leading to its ubiquitination and degradation by the proteasome, which decreases the NRF2-mediated antioxidant pathway in ECs. The NRF2 pathway can be reactivated by the use of the NRF2 activator tBHQ, as depicted in the model (Fig. 2h).

**Fig. 1 | TRIM47 is an activator of the antioxidant NRF2 pathway in human brain endothelial cells. a** RNA-sequencing performed on HBMEC treated with control or TRIM47 siRNA for 72 h. Volcano plot showing log2 fold change (FC) vs −log10 FDR of differentially expressed genes (DEG) in response to TRIM47 knockdown. Significantly downregulated (log2FC < −1, in blue, 208 genes) and upregulated (log2FC > 1, in red, 214 genes) genes were identified (FDR < 0.05 and CPM > = 5). **b** Pathway enrichment analysis of DEG (Wikipathway) revealed a significant (p-adj = 0.025) downregulation of genes associated with the NRF2 signaling pathway. The dotplot depicted the normalized enrichment scores (NES) for the top 10 pathways. **c** Heatmap showing the expression of selected genes related to the NRF2 pathway in HBMEC treated with TRIM47 siRNA vs control. Data are expressed as log2FC. *FDR < 0.05. **d** Immunoblots and quantification of WB for TRIM47 and HO1 expression in control and TRIM47-deficient HBMEC treated with DMSO or MG132 for 5 h. Protein lysates from the same samples were loaded onto two separate gels and analyzed independently. Data were normalized to α-tubulin (TRIM47) or beta actin (HO1). ($n$ = 3 or 4 experiments). ****$P$ < 0.0001, Student's $t$ test. **e, f** qPCR analysis of (**e**) *HMOX1* and (**f**) *NQO1* gene expression in control, TRIM47 or NRF2 siRNA-treated HBMEC for 72 h and treated with either PBS or H2O2 (200 μM for 3 h). Data were normalized to *cyclophilin* ($n$ = 3 experiments). *$P$ < 0.05; **$P$ < 0.01; ***$P$ < 0.001, One-way ANOVA, #$P$ < 0.05; ##$P$ < 0.01, Student's $t$ test. **g** HO1 promoter luciferase reporter assay. HeLa were co-transfected with HO1 promoter-luciferase construct (HO1−4.5 bp) or a pGL3 empty vector with control or siRNA targeting TRIM47. Cells were treated with either PBS or H2O2 (200 μM for 1 h). Luciferase activity was measured in cell lysates. Values represent the fold change in relative luciferase activity over the HO1-4.5 kb + siControl condition and normalized to Renilla ($n$ = 4 experiments). ****$P$ < 0.0001 One-way ANOVA; #### $P$ < 0.0001 compared to pGL3 + siControl condition, Student's $t$ test. All graphical data are mean ± s.d.

## TRIM47 protects from oxidative stress both in vitro and in vivo

Since TRIM47 promotes the antioxidant NRF2 pathway, we hypothesized that loss of TRIM47 in brain ECs might lead to a higher susceptibility to stress. Indeed, depletion of *TRIM47* in HBMEC led to a significant increase in OS as detected by a CellRox green dye. Additionally, *TRIM47* siRNA worsened the effect of two OS inducers, tert-butyl hydroperoxide (TBHP) and Angiotensin II (Supplementary Fig. 4a, b). Overexpression of TRIM47 prevented the ROS production induced by TRIM47 siRNA, demonstrating that the increased OS was triggered by the loss of TRIM47 (Supplementary Fig. 4c, d). Notably, activation of the NRF2 pathway by stabilizing NRF2 protein with the tBHQ compound was able to dampen the OS induced by *TRIM47* depletion under both basal and stress conditions (Fig. 3a, b). This indicates that TRIM47 contributes to antioxidant defenses in HBMEC, at least in part by activating the NRF2 pathway.

To explore the antioxidant potential of TRIM47 in vivo, we used mice deleted for the *Trim47* gene across all tissues and cell types (knock out (KO) mice also denoted *Trim47−/−* mice) (Supplementary Fig. 5a). *Trim47−/−* mice were viable and fertile (Supplementary Fig. 5b), developed normally, showing no significant difference in weight compared to their littermate controls (*Trim47+/+* mice) (Supplementary Fig. 5c–e) and displaying no obvious macroscopic phenotype. The efficacy of *Trim47* deletion was confirmed at the mRNA level by qPCR on brain lysates and isolated brain ECs from *Trim47−/−* and *Trim47+/+* mice (Fig. 3c, d). Endothelial enrichment using our isolation protocol was verified by qPCR analysis. The results showed a significant increase in endothelial markers, along with a moderate enrichment of pericyte markers, and a marked reduction in neuronal (*Tuj1*) and myelin (*S100b*) markers (Supplementary Fig. 5f, g).

To challenge these KO mice at a postnatal age, we employed a pathological model of oxidative stress induced by hypoxia, namely the oxygen-induced retinopathy (OIR) model, which is characterized by increased reactive oxygen species (ROS) production[41,42]. qPCR analysis of *Nfe2l2* and its target genes in brain tissues from *Trim47+/+* control mice exposed to either room air or the OIR model, confirmed that this pathological model activates the NRF2 protective antioxidant system in the central nervous system (Fig. 3c). Notably, deletion of *Trim47* resulted in a decrease in the *Nrf2* pathway activation under both baseline and pathological (OIR) conditions (Fig. 3c). A similar reduction in Nrf2 pathway activation was observed in brain ECs isolated from *Trim47+/+* and *Trim47−/−* mice following exposure to the OIR model (Fig. 3d). Additionally, histological analysis of the retinas at p16 after exposure to OS indicated that the loss of *Trim47* was associated with an increase in avascular areas (Fig. 3e, f), suggesting that Trim47 plays a vasculoprotective role during the hyperoxia-induced capillary vaso-obliteration phase. To investigate the potential contribution of Trim47 to physiological angiogenesis that might underlie this phenotype, we collected retinas at postnatal stages p6 and p12. Analysis of the vascular network at p6 revealed no angiogenic defects in *Trim47*-deficient retinas (Supplementary Fig. 6a–e). These findings were confirmed at p12 (Supplementary Fig. 6f, g), demonstrating that Trim47 is dispensable for postnatal angiogenesis in the mouse retina.

Together, these results demonstrate that TRIM47 provides antioxidant properties to brain ECs in vitro and protects the vasculature in a postnatal mouse model (OIR) marked by ROS production and inflammation, at least partly by promoting the Nrf2 pathway.

## *Trim47* deletion leads to cognitive impairment

Recent large-scale genomic analyses in humans have identified *TRIM47* as a putative causal gene for common, multifactorial cSVD[21]. To investigate the impact of *Trim47* deletion on cognitive functions and brain physiology, we submitted adult *Trim47−/−* mice and their control littermates to two complementary behavioral tests assessing spatial recognition memory in the Y-maze and hippocampus-dependent spatial discrimination in the Morris water maze. Contrary to controls, male *Trim47−/−* mice (Fig. 4a, b) failed to recognize the unexplored (previously inaccessible) arm during the test phase of the Y-maze procedure, indicating an inability to form short-term spatial recognition memory. Adult *Trim47−/−* mice were also impaired in the cognitively-challenging water maze paradigm (Fig. 4c–g) where they were unable to learn the spatial location of the hidden platform. Accordingly, latency and distance swum to the platform decreased over the 4 training days in control but not in *Trim47−/−* mice (Fig. 4c–e). Potential confounders such as locomotor deficits were absent in the *Trim47−/−* mice whose swim speed was comparable to that of controls (Fig. 4f). Cognitive impairments were similar in female cohorts, suggesting that the phenotype observed in *Trim47*-deficient mice is not gender-specific (Fig. 4h–n). Impaired search accuracy in *Trim47* mutants was confirmed during a probe test without the platform given 3 days following training completion. The highly sensitive proximity measure to the former platform location[43] was lower in both *Trim47−/−* males and females compared to their respective control littermates (Fig. 4e, l and Supplementary Fig. 7a–f). To gain further insights into the spatial memory deficit exhibited by *Trim47* mutants, we next analyzed navigation strategies adopted by *Trim47+/+* and *Trim47−/−* mice during training using a previously reported search strategy classification[44]. On the first day of acquisition training, regardless of sex, both controls and *Trim47* mutants used a comparable mix of spatial (direct) and nonspatial (circling or random) swim strategies (Supplementary Fig. 7g, h). As training progressed, control littermates switched toward a predominant spatial strategy, a contribution likely responsible for their improved accuracy in locating the hidden platform. In contrast, *Trim47* mutants persisted in using predominantly nonspatial, hippocampal-independent strategies, potentially indicating an alteration in hippocampal functions associated with the *Trim47* mutation (Supplementary Fig. 7g, h). Together, these data indicate that the loss of *Trim47* is associated with impairment in spatial hippocampal-dependent memory.

Interestingly, the sucrose preference test did not reveal any sign of anhedonia, a key symptom of depression in humans, suggesting that *Trim47−/−* mice develop a purely cognitive impairment without any apparent anxiety-like symptoms (Supplementary Fig. 7i).

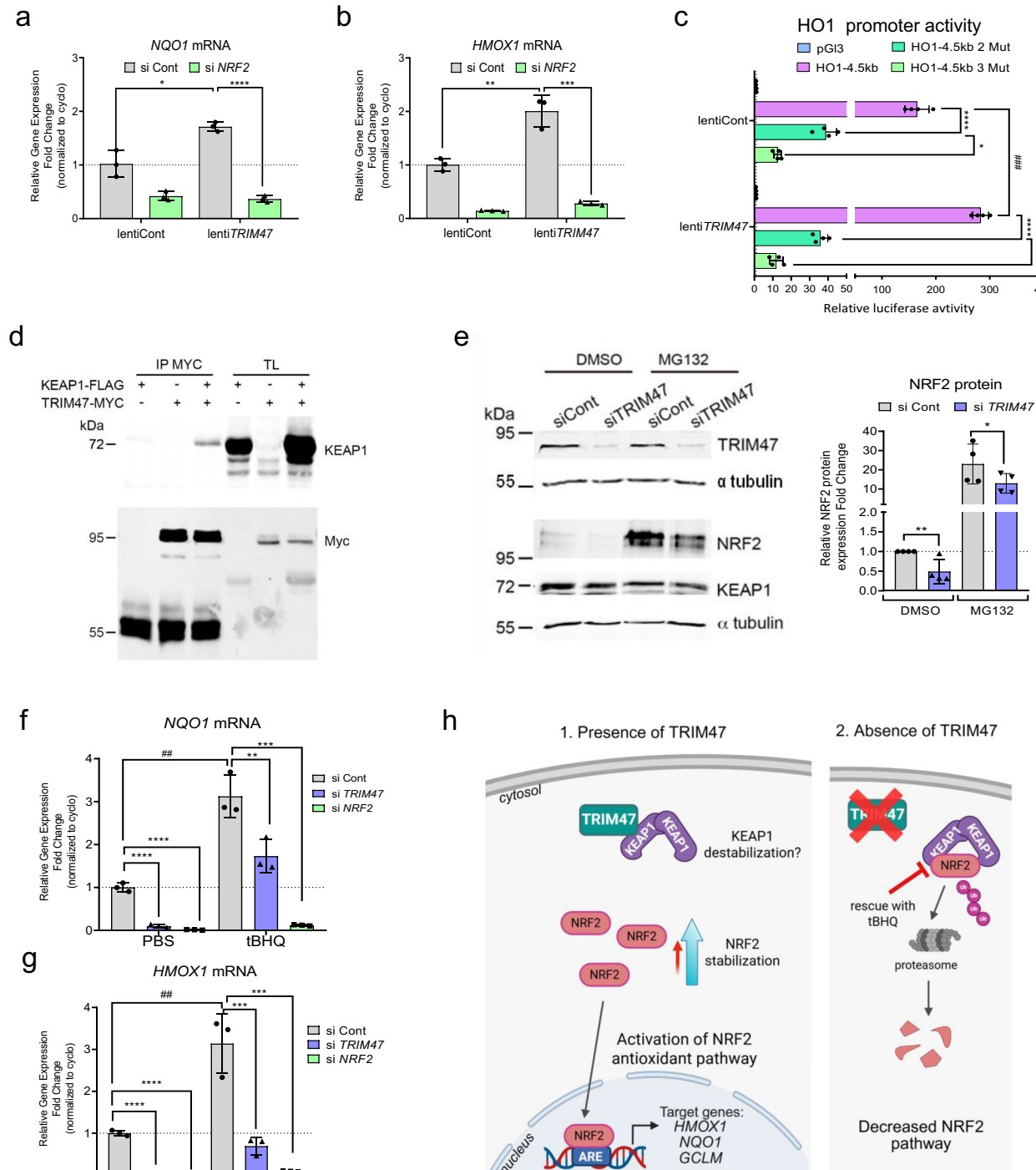

**Fig. 2 | TRIM47 cooperates with NRF2 in driving antioxidant target genes.**
**a**, **b** qPCR analysis of (**a**) *NQO1* and (**b**) *HMOX1* gene expression in siCont or siNRF2 treated HBMEC and co-transduced with a control or TRIM47 lentivirus for 24 h. Data were normalized to *cyclophilin* (*n* = 3 experiments). *P < 0.05; **P < 0.01; ***P < 0.001; ****P < 0.0001, One-way ANOVA. **c** HO1 promoter luciferase reporter assay. TRIM47 cDNA expression plasmid (TRIM47) or empty expression plasmid (pcDNA) were co-transfected with HO1 promoter-luciferase constructs (HO1-4.5 bp) or a pGL3 empty vector in HeLa, and luciferase activity was measured. Values represent the fold change in relative luciferase activity over pGL3 vector (*n* = 4 experiments). *P < 0.05; **P < 0.01; ****P < 0.0001, One-way ANOVA; ### P < 0.001, Student's *t* test. **d** TRIM47 and KEAP1 interaction was assessed by Co-IP assay in whole cell lysates from HEK293. Cells were transfected with TRIM47-myc pcDNA, KEAP1-Flag or both expression plasmids. Lysates were immunoprecipitated with myc antibody and immuno-blotted for myc (TRIM) and KEAP1. **e** Immunoblots and quantification of WB for NRF2 and KEAP1

in control and TRIM47-deficient HBMEC treated with DMSO or MG132 for 5 h (*n* = 4 experiments). Protein lysates from the same samples were loaded onto two separate gels and analyzed independently. Data were normalized to α-tubulin.*P < 0.05; **P < 0.01, Student's *t* test. **f**, **g** qPCR analysis of (**f**) *NQO1* and (**g**) *HMOX1* gene expression in HBMEC treated with control, TRIM47 or NRF2 siRNA for 72 h and with tBHQ (10 μM) for 24 h (*n* = 3 experiments). **P < 0.01; ***P < 0.001; ****P < 0.0001, One-way ANOVA; ##P < 0.01 Student's *t* test. All graphical data are mean ± s.d. **h** TRIM47 activation of the antioxidant NRF2 pathway. 1. In basal conditions, TRIM47 binds to KEAP1 in the cytoplasm and thus alleviates KEAP1-dependent NRF2 degradation and increases NRF2 protein stabilization that can drive the expression of NRF2-target genes (*HMOX1*, *NQO1*) promoting cellular antioxidant protection. 2. In the absence of TRIM47, KEAP1 strongly binds to NRF2, leading to its ubiquitination and degradation by the proteosomal complex, thus leading to the decrease in the NRF2 pathway. Created in BioRender (2026) https://BioRender.com/zl4bser.

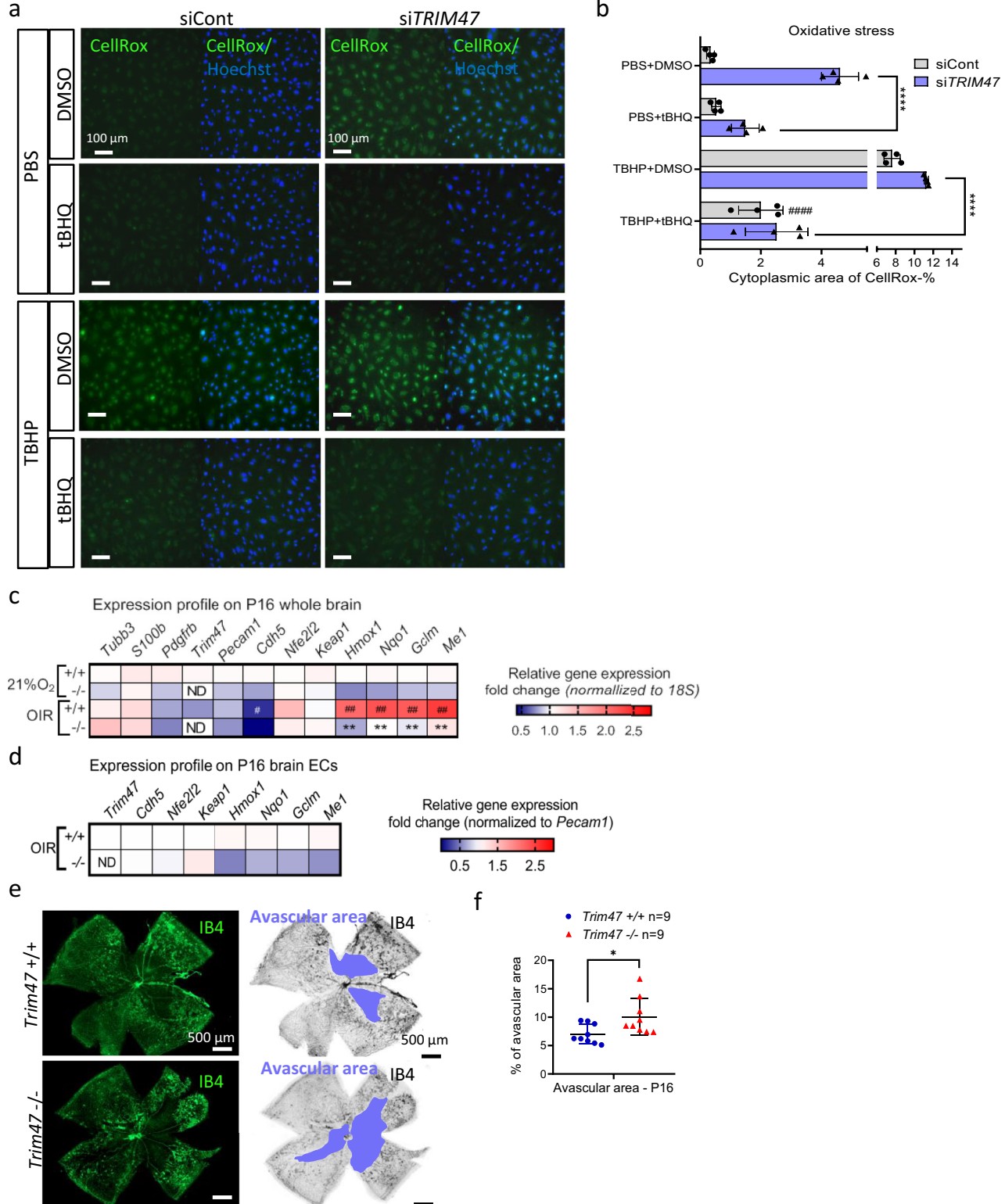

### Trim47 deletion triggers BBB dysfunction and decreased Nrf2 pathway activation

The enrichment of *Trim47* expression in mouse brain ECs and preliminary findings from cell-based models showing enhanced endothelial permeability following TRIM47 silencing in vitro[21] pointed towards its potential role in blood–brain barrier (BBB) development and/or function. Notably, Glut1 immunostaining analysis in brain sections from adult *Trim47−/−* and control mice showed no differences in vascular density in the cortical regions and

hippocampus compared to control mice (Supplementary Fig. 8a, b). These data prove that *Trim47* is not essential for the development of brain blood vessels.

To determine whether the cognitive deficits observed in *Trim47* KO mice were linked to compromised BBB function, we assessed BBB integrity by injecting two different tracers. Quantification of cadaverine (Fig. 5a and Supplementary Fig. 8c) and dextran (Fig. 5b and Supplementary Fig. 8d) extravasation, expressed as the permeability index ratio, revealed a significant increase in BBB leakiness in adult *Trim47−/−* mice compared to

**Fig. 3 | TRIM47 protects from oxidative stress in vitro and in vivo.**
**a, b** Representative immunofluorescence image (**a**) and quantification (**b**) of CellRox dye (green, marker of oxidative stress) in HBMEC transfected with siCont or si*TRIM47* for 24 h, then pre-treated with DMSO or tBHQ (10 µM for 24 h) and treated with PBS or tert-butyl hydroperoxide (TBHP, 200 µM for 1 h); nuclei are identified by Hoechst (blue). Scale bar 100 µm. Quantification represents the mean percentage of cytoplasmic area of Cellrox dye per field ($n = 4$ wells from 3 independent experiments). ****$P < 0.0001$, One-way ANOVA; ####$P < 0.001$: comparison with siCont + THBP without drug treatment. **c** Heatmaps of selected genes link to the Nrf2 pathway. qPCR screening performed on whole brain lysates isolated from P16 *Trim47+/+* and *Trim47−/−* mice exposed to normoxia condition or oxygen-induced retinopathy (OIR) model and. Data were normalized to *18S* ($n = 4$–7 mice per genotype and condition). #$P < 0.05$, ##$P < 0.01$, +/+ OIR *vs* +/+ 21%O$_2$; Mann–Whitney test. **$P < 0.01$, −/− OIR *vs* +/+ OIR; Mann–Whitney test. ND: not detectable. **d** qPCR in brain ECs isolated from P16 *Trim47+/+* and *Trim47−/−* mice exposed to OIR model. Data were normalized to *Pecam1* and expressed as fold change ($n = 3$ replicates/ genotype; 2 mice/replicate). **e** Representative images of isolectin B4 (IB4, green or black) staining of postnatal day 16 retina from *Trim47+/+* and *Trim47−/−* mice following OIR model characterized by oxidative stress. The avascular area of the retina after OIR is highlighted in blue. Scale bar 500 µm. **f** Quantification represents the percentage of avascular area (µm$^2$) in *Trim47+/+* and *Trim47−/−* mice at P16 following OIR ($n = 9$ mice per genotype). *$P < 0.05$; Mann–Whitney test. All graphical data are mean ± s.d.

control mice. Importantly, tracer permeability in the kidney was not affected, suggesting that *Trim47* specifically acts on brain vasculature which is characterized by a specialized continuous endothelium (Fig. 5a, b). Immunostaining of brain sections with fibrinogen and podocalyxin antibodies, to visualize blood vessels, revealed the presence of fibrinogen, a blood component, outside the vessels in various brain regions including in the hippocampus, and cortical regions of *Trim47−/−* mice (Fig. 5c and Supplementary Fig. 9a, b). In contrast, fibrinogen was confined within the vessels in control mice, confirming a localized increase in BBB permeability (in some capillaries and larger vessels) in *Trim47−/−* mice (Fig. 5c).

To identify the cause of BBB defects, we analyzed the expression of key elements of tight and adherent junctions which are essential for BBB integrity, in adult *Trim47−/−* and *Trim47+/+* mice. qPCR analysis of brain ECs (Fig. 5d) showed a mild but significant decrease in the expression of the tight junction components *Cldn5* (Claudin5), *Cldn11* (Claudin11), and *Ocln* (Occludin) in *Trim47−/−* mice compared to control mice (Fig. 5d). However, the mRNA expression level of *Tjp1* (ZO1) was not affected in *Trim47−/−* mice compared to their littermate controls (Fig. 5d). Additionally, there was a trend towards a downregulation of *Cdh5* (VE-cadherin, a major element of adherens junction) in both brain ECs (Fig. 5d) and in whole brain lysates (Supplementary Fig. 8e). Importantly, immunofluorescence staining of brain sections confirmed the reduced protein levels of Claudin5 in the hippocampus and cortical regions of *Trim47−/−* mice (Fig. 5e and Supplementary Fig. 10). These data indicated that the increased BBB permeability observed in *Trim47−/−* mice is likely due to the decreased expression of specific tight/adherent junction components in brain ECs.

qPCR analysis confirmed the downregulation of the Nrf2 pathway activation in adult *Trim47*-deficient mice, both in brain EC (Fig. 5f) and the whole brain (Supplementary Fig. 8e). These data were confirmed at the protein level by immunostaining of brain tissues, which showed significantly reduced HO-1 expression in the vasculature of *Trim47−/−* adult mice (Supplementary Fig. 11). These findings align with our in vitro and in vivo data from the hypoxia-induced OS model (OIR) at postnatal age. Blood vessels are particularly vulnerable to localized OS, and reduced antioxidant protection in ECs may further contribute to BBB impairment. Our results demonstrate that the loss of BBB integrity in *Trim47−/−* mice is associated with decreased expression of tight junction proteins. Nrf2 has been reported as a positive transcriptional regulator of Claudin5 expression in ECs[45], suggesting that the decreased Nrf2 pathway in *Trim47*-deficient brain ECs could aggravate damage through the downregulation of Claudin5, thereby directly contributing to BBB dysfunction.

**BBB impairment induced by Trim47 deletion leads to brain lesions in vivo**

Since BBB disruption can lead to the leakage and accumulation of neurotoxic material or blood components (such as fibrinogen) in the brain parenchyma, we next wanted to determine whether BBB impairment in *Trim47−/−* mice resulted in additional brain defects. Astrocytes, which are in close contact with ECs, are highly plastic and can be directly affected by BBB dysfunction and changes in their local environment[46,47]. Supporting this idea, GFAP staining indicated a significant increase in astrocyte activation in the hippocampus, particularly in the dentate gyrus region of *Trim47−/−* mice where blood vessel leakiness was observed (Fig. 5g).

As astrocyte reactivity can be modulated by neuroinflammation[46], we next assessed the expression of adhesion molecules and cytokines in brain ECs and in whole brain lysates. qPCR analysis revealed no clear evidence of a pro-inflammatory endothelial state, as indicated by a trend toward reduced expression of the adhesion molecules *Icam1* and *Vcam1* (Fig. 5d and Supplementary Fig. 8e), along with no marked increase in cytokine expression (*Il1β*, *Il6*, and *Tnfα*) or dysregulation of the canonical interferon pathway in *Trim47−/−* brain ECs (Fig. 5d). To further assess immune activation, we analyzed immune cell and microglial markers in *Trim47+/+* and *Trim47−/−* brain samples by measuring *Cd11b* mRNA expression levels (Supplementary Fig. 12a) and by performing immunofluorescence staining with Iba1 (Supplementary Fig. 12b) and Cd68 (Supplementary Fig. 12c) antibodies. Collectively, these findings indicated that *Trim47* deletion does not lead to immune cell infiltration or marked activation of microglia and perivascular macrophages at the time of brain collection (i.e., in 7–10-month-old mice), suggesting a possible pre-inflammatory stage in middle-aged *Trim47*-deficient mice.

Since *Trim47* mutant mice show cognitive impairment, we hypothesized that the loss of *Trim47* may affect the number of viable neurons, axonal density and/or myelination. However, histological analyses of coronal brain sections revealed no evidence of detectable neuronal loss in *Trim47−/−* mice compared to controls, as shown by NeuN immunostaining in the hippocampus and cortical regions (Supplementary Fig. 12d, e). GWAS have identified the *TRIM47* locus as a lead genetic risk locus for high burden of WMH in humans[48]. To evaluate the impact of *Trim47* deletion on brain WM, we performed myelin basic protein (MBP) staining on brain tissues. Analysis of MBP intensity and area did not show any significant reduction in myelination in the hippocampus of 10-month-old *Trim47−/−* mice compared with *Trim47+/+* mice (Fig. 5h), suggesting that *Trim47*-deficient mice do not display major detectable WM lesions at this stage in the hippocampus region.

**Endothelial-specific deletion of *Trim47* recapitulates the vascular-dementia phenotype**

To demonstrate that the phenotype observed in *Trim47−/−* mice was primarily due to the loss of endothelial *Trim47*, we generated mice deleted for *Trim47* in ECs (*Trim47*iEC-KO) and compared them with littermate controls (*Trim47*iEC-WT) using an inducible CDH5-Cre line (Supplementary Fig. 13a–d). The efficiency of *Trim47* deletion was confirmed on lung ECs isolated from *Trim47*iEC-KO versus *Trim47*iEC-WT adult mice (Fig. 6a). Behavioral tests conducted on adult mice revealed that male *Trim47*iEC-KO mice displayed impaired performance in the Y-maze (Fig. 6b, c) and water maze procedures (Fig. 6d–h) similar to the deficits observed in *Trim47−/−* mice. These findings confirm that the loss of endothelial *Trim47* is sufficient to induce cognitive impairment.

Importantly, the endothelial-specific deletion of *Trim47* also led to impaired BBB function, as shown by an increased BBB permeability index following 3 kDa dextran injection (Fig. 6i) and a decreased expression of key adherent and tight junction components in brain ECs isolated from these mice (Fig. 6k). Finally, qPCR analysis also highlighted the downregulation of critical Nrf2 target genes, including *Hmox1* and *Slc7a11*, in brain ECs from *Trim47*iEC-KO compared to *Trim47*iEC-WT controls (Fig. 6k).

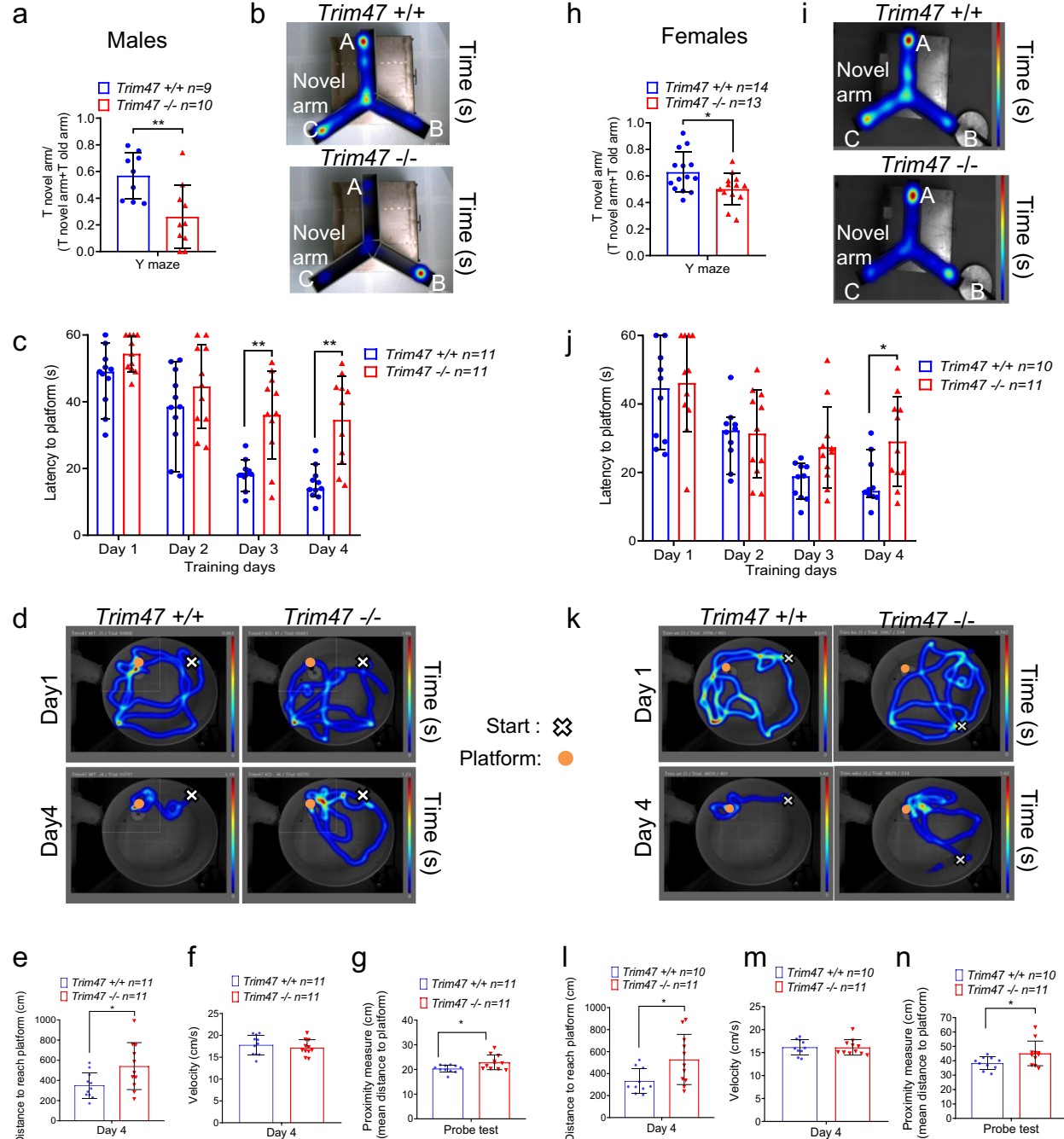

**Fig. 4 | Trim47 deletion leads to cognitive impairment. a**, **b** Y-maze test was performed on males (8–9 months; *Trim47+/+*: *n* = 9; *Trim47−/−*: *n* = 10) to assess recognition memory in *Trim47+/+* and *Trim47−/−* mice. **a** Y-maze data are presented as a ratio between the time spent in novel arm and the cumulative time spent in the novel and the familiar arms. **\*\****p* < 0.01, −/− vs +/+ mice, Mann–Whitney test. **b** Representative heatmap images of the paths adopted in the Y-maze test. Each color represents how long the mouse stayed in that location (time in seconds). Warmer colors (red/yellow) = more time spent. Cooler colors (blue) = less time spent. Mice started in position A; novel arm is arm C. **c–g** Water maze test was performed in adults male (7–8 months) *Trim47+/+* (*n* = 11) *and Trim47−/−* (*n* = 11) mice and revealed lower performances. **c** Graph shows the latency (time) to reach the hidden platform (expressed in seconds) over 4 training days. **\*\****p* < 0.01 −/− vs +/+ mice, Two-way ANOVA with repeated measures. **d** Representative heatmap images the paths taken by the mice and, showing their learning progression from the first to the last day of the training session. Each color represents how long

the mouse stayed in that location (time in seconds). Warmer colors (red/yellow) = more time spent. Cooler colors (blue) = less time spent. The orange circle shows the position where the hidden platform was located; the white cross indicates where the mice started the test. (e) Graph shows the distance to reach platform (expressed in cm) on day 4. **\****p* < 0.05, −/− vs +/+ mice, Mann-Whitney test. (f) Data represent the velocity expressed as cm per second of each mouse on day 4 of water maze training. **g** Graph shows the proximity measures (in cm) to platform during the probe test performed 72 h after the training. **\****p* < 0.05, −/−vs +/+ mice, Mann–Whitney test. **h–n** Y-maze was also performed in females (8–10 months; *Trim47+/+*: *n* = 14; *Trim47−/−*: *n* = 13). **\****p* < 0.05, −/− vs +/+ mice, Mann–Whitney test. **i–l** Water maze test was repeated using females (8–9 months, *Trim47+/+*: *n* = 10; *Trim47−/−*: *n* = 11). **j** **\****p* < 0.05, −/− vs +/+ mice, Two-way ANOVA. **l**, **n** **\****p* < 0.05, −/− vs +/+ mice, Mann–Whitney test. All graphical data are mean ± s.d.

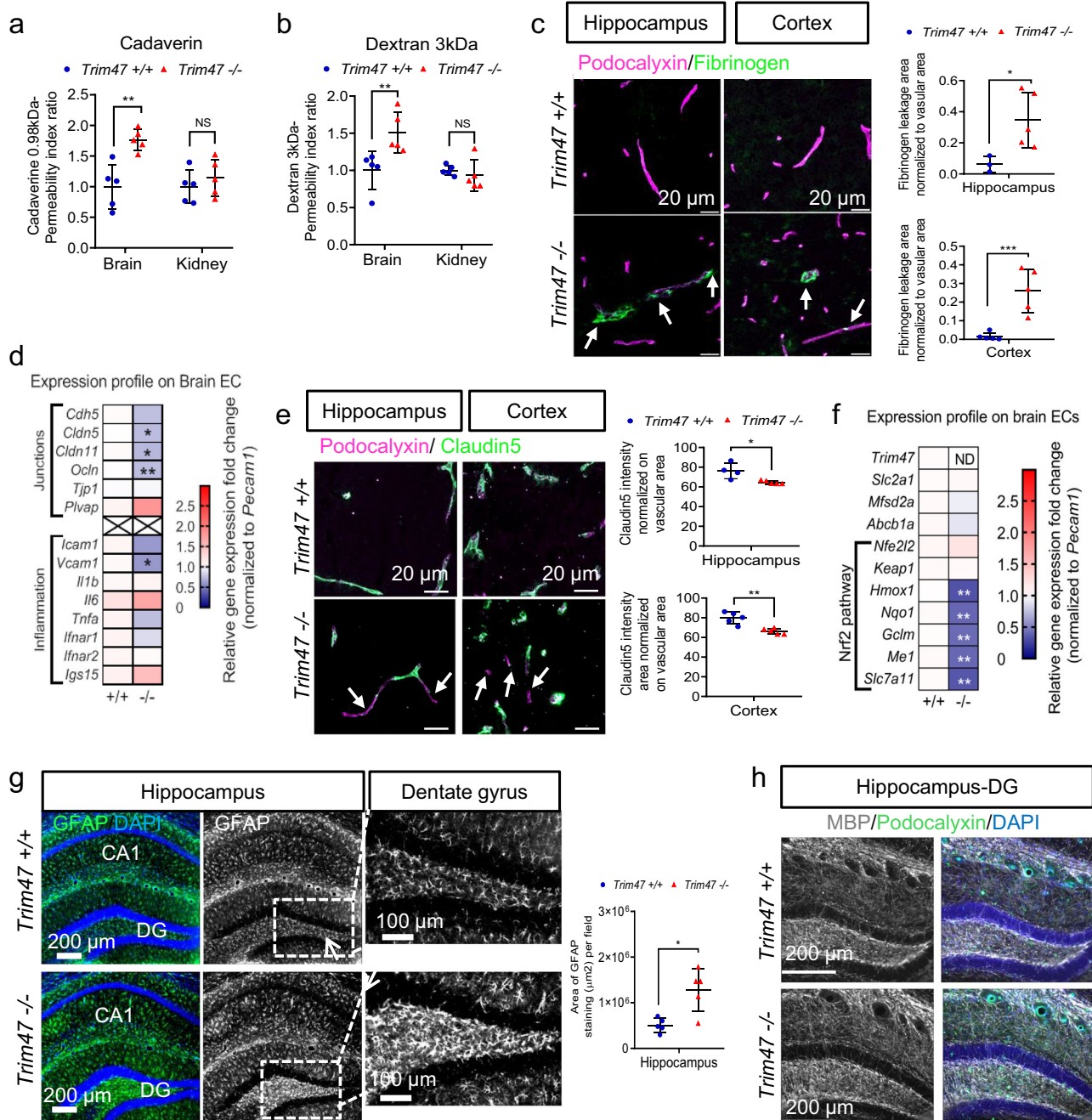

**Fig. 5 | Trim47 deletion leads to brain defects associated with Nrf2 pathway impairment. a–c** Blood–brain barrier permeability assessment was done on *Trim47+/+* and *Trim47−/−* mice (5–7 months, males). Quantification of (**a**) cadaverine and (**b**) dextran contents in brain and kidney. Data indicate the permeability index ratio (*n* = 5 *Trim47+/+* and n = 5 *Trim47−/−*).NS: not significant, *P* > 0.05; **P < 0.01, Mann–Whitney test. **c** Representative images and quantification of fibrinogen leakage (green) in brain cryosections from *Trim47+/+* and *Trim47−/−* mice. Tissues are co-stained for podocalyxin (blood vessels, pink). Scale bars 20 μm. White arrows highlight fibrinogen extravasation in *Trim47−/−* mice. Graphs represent permeability in hippocampus (CA1) (*n* = 3 *Trim47+/+*, *n* = 5 *Trim47−/−*) and in cortex (*n* = 5 *Trim47+/+*, *n* = 5 *Trim47−/−*). *P < 0.05; ***P < 0.001, Mann–Whitney. **d** qPCR analysis of selected genes in brain ECs isolated from *Trim47+/+* and *Trim47−/−* mice (7–10 months). Data normalized to *Pecam1* (*n* = 6 replicates/genotype, except for *Il1b, Il6* and *Plvap*: *n* = 3/genotype, 2 mice/replicate). ND: not detectable; *P < 0.05; **P < 0.01, Mann–Whitney.

**e** Representative images and quantification of claudin5 (green) in brain cryosections from *Trim47+/+* and *Trim47−/−* mice (6–7 months). White arrows highlight blood vessels areas without claudin5 expression. Scale bars 20 μm. Graphs show claudin5 expression in hippocampus (CA1) (*n* = 4 *Trim47+/+*, *n* = 5 *Trim47−/−*) and in cortex (*n* = 5 per genotype). *P < 0.05; **P < 0.01, Mann–Whitney. **f** qPCR for Nrf2 target genes in brain EC from *Trim47+/+* and *Trim47−/−* mice (7–10 months). Data normalized to *Pecam1* (*n* = 6 replicates/genotype, 2 mice/replicate; **P < 0.01, Mann–Whitney. ND: not detectable. **g** Representative images and quantification of GFAP (green/white) in brain sections (hippocampus) from *Trim47+/+* and *Trim47−/−* mice (10 months) (*n* = 5 *Trim47+/+* and *n* = 5 *Trim47−/−*). Scale bars 200 or 100 μm. *P < 0.05, Mann–Whitney. **h** Representative image of MBP (white) staining in brain sections (hippocampus) from *Trim47+/+* and *Trim47−/−* mice (8 months). DG: gyrus dentate. Scale bars 100 μm. Graphical data are mean ± s.d.

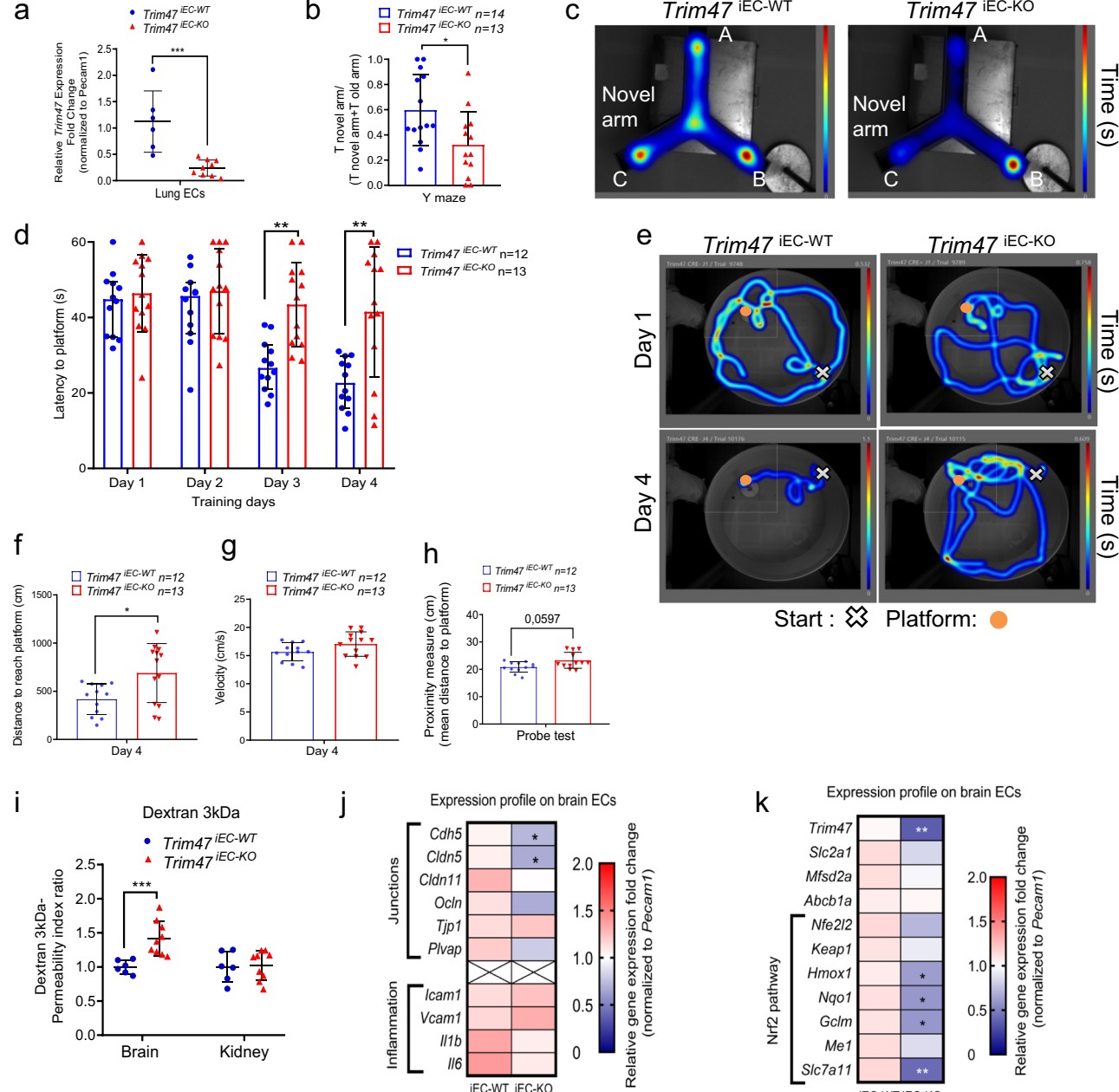

**Fig. 6 | Trim47 deletion in endothelial cells is sufficient to mimic the phenotype observed in Trim47 KO mice. a** qPCR analysis for Trim47 expression was done on lung ECs isolated from Trim47[iEC-WT] and Trim47[iEC-KO] males (5 months). Data normalized to Pecam1 ($n = 6$ Trim47 [iEC-WT] and $n = 9$ Trim47 [iEC-KO] mice); ***$P < 0.001$, Mann–Whitney. **b**, **c** Y-maze test was performed on males (3–5 months; Trim47[iEC-WT]: $n = 14$; Trim47 [iEC-KO]. $n = 13$). **b** Y-maze data are presented as a ratio between the time spent in novel arm and the cumulative time spent in the novel and the familiar arms. *$p < 0.05$, Mann–Whitney test. **c** Representative heatmap images of the path for the Y-maze test. Mice started in position A; novel arm is arm C. **d**–**h** Water maze test was performed on males (3–5 months; Trim47 [iEC-WT]: $n = 12$; Trim47[iEC-KO]: $n = 13$). **d** Graph shows the latency (time) to reach the hidden platform (expressed in seconds) for the 4 training days. **$p < 0.01$, iEC-WT vs iEC-KO mice, Two-way ANOVA with repeated measures. **e** Representative heatmap images of the paths for the first and last training days. **f** Graph shows the

distance to reach platform (in cm) on day 4. *$p < 0.05$, iEC-WT vs iEC-KO mice, Mann–Whitney test. **g** Data represent the velocity expressed as cm per second (swimming speed) of each mouse on day 4 during water maze training. **h** Graph shows the proximity measures (in cm) to platform during the probe test performed 72 h after the training. *$p < 0.05$, iEC-WT vs iEC-KO mice, Mann–Whitney test. **i** Assessment of BBB permeability was done on Trim47[iEC-WT] and Trim47[iEC-KO] mice (4–6 months, males). Quantification of dextran contents in brain cortex and kidney, 20 min after I.V injection of fluorescent tracer. Data indicate the permeability index ratio (normalization to fluorescence detected in serum) ($n = 5$ Trim47[iEC-WT] and $n = 5$ Trim47[iEC-KO]). ***$P < 0.001$, Mann–Whitney test. **j**, **k** qPCR for (**j**) selected genes and (**k**) Nrf2 targets in brain EC isolated from Trim47[iEC-WT] and Trim47[iEC-KO] mice (4–6 months). Data normalized to Pecam1 ($n = 4$–6 replicates/genotype, 2 mice/replicate; *$P < 0.05$; **$P < 0.01$, Mann–Whitney. All graphical data are mean ± s.d.

These results demonstrate that the loss of *Trim47* specifically in ECs recapitulates the vascular cognitive impairment phenotype observed in the global KO mouse, underscoring the crucial role of endothelial Trim47 in maintaining brain homeostasis.

## Activation of the NRF2 pathway prevents brain defects and cognitive impairment

Since we showed that TRIM47 is a key regulator of the NRF2 antioxidant pathway both in vitro and in vivo, we hypothesized that *Trim47* could

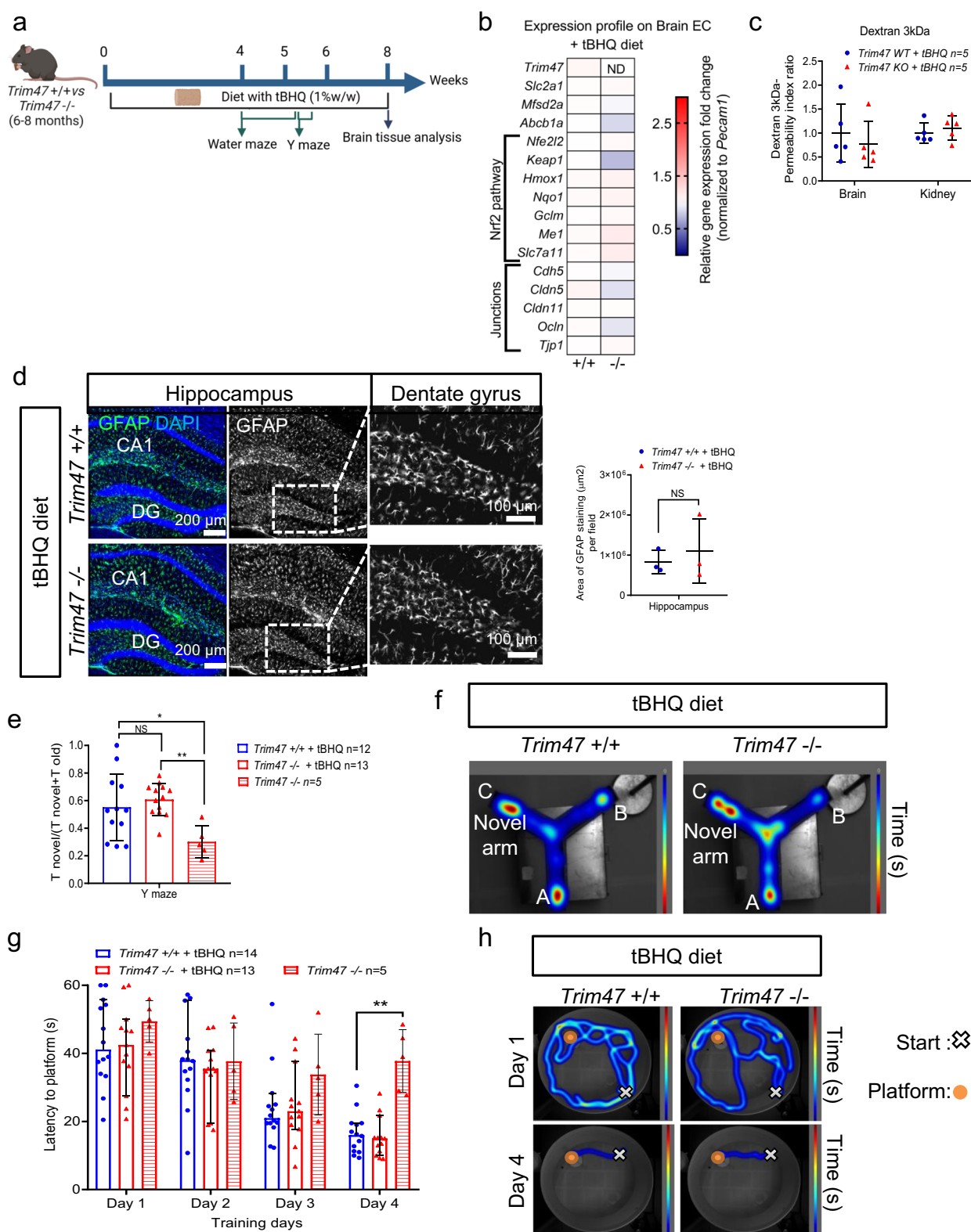

promote brain homeostasis by activating the Nrf2 protective pathway in adult mice. To test this, we performed experiments on male *Trim47+/+* and *Trim47−/−* mice, feeding them a diet containing the Nrf2 activator tBHQ (1% w/w) (Fig. 7a). As previously reported[49,50], the tBHQ diet was well tolerated by the mice and did not induce any weight loss (Supplementary Fig. 14a). qPCR analysis confirmed that the tBHQ diet effectively prevented

the impairment of the antioxidant pathway in *Trim47−/−* mice, both in brain ECs (Fig. 7b) and whole brain tissue (Supplementary Fig. 14b).

Notably, the expression of tight junction components in brain ECs was restored to normal levels in *Trim47−/−* mice treated with the Nrf2 activator compared to control mice (Fig. 7b). The diet with the antioxidant molecule prevented the increased BBB permeability (Fig. 7c), proving that antioxidant

**Fig. 7 | Activation of NRF2 pathway prevents cognitive impairment and brain defects observed in Trim47 KO mice. a** Experimental therapeutical approach chosen in mouse. 6–8 months *Trim47+/+ and Trim47−/−* males were placed on a pellet diet containing the NRF2 pathway activator tert-butylhydroquinone (tBHQ). Behavioral tests were then performed on these mice and on untreated *Trim47−/−* males before brain harvesting for qPCR analysis, BBB function and histology. Created in BioRender (2026) https://BioRender.com/cbhgf8u. **b** qPCR for Nrf2 pathway performed on brain EC from *Trim47+/+ and Trim47−/−* males (8–10 months) with tBHQ diet ($n = 5$ replicates/genotype, 2 mice/replicate). ND: not detectable. **c** BBB permeability assessment done on *Trim47+/+ and Trim47−/−* males with tBHQ (8-9 months). Quantification of dextran in brain cortex and kidney ($n = 5$ *Trim47+/+* + tBHQ and $n = 5$ *Trim47−/−* +tBHQ). Mann–Whitney. **d** Representative image and quantification of GFAP expression (green) in brain sections (coronal) from *Trim47+/+* and *Trim47−/−* adult mice (8–10 months) after tBHQ diet ($n = 3$ per genotype). Scale bars 100 μm. NS: not significant, $P > 0.05$, Mann–Whitney. **e, f** Y-maze test was performed on adult mice after a 1 month diet with tBHQ. **e** Data are presented as a ratio between the time spent in novel arm and the cumulative time spent in the novel and familiar arm ($n = 12$ *Trim47+/+* + tBHQ, $n = 13$ *Trim47−/−* + tBHQ and $n = 5$ *Trim47−/−* without diet). NS: $P > 0.05$, $*p < 0.05$; $**p < 0.01$, One-way ANOVA. **f** Representative heatmap images of the paths for the Y-maze test. **g, h** Water maze test performed on adult mice after tBHQ diet. **g** Latency to reach platform (expressed in seconds) is shown for the 4 training days ($n = 14$ *Trim47+/+* + tBHQ, $n = 13$ *Trim47−/−* + tBHQ, $n = $ *Trim47−/−*). $*p < 0.05$, *Trim47−/−* vs *Trim47+/+* + tBHQ. Two-way ANOVA with repeated measures. **h** Representative heatmap of the paths of *Trim47 +/+ and Trim47 +/+* treated with tBHQ. All graphical data are mean ± s.d.

therapy successfully restored BBB integrity and function in *Trim47*-deficient mice.

In line with its known anti-inflammatory properties[51,52], tBHQ did not normalize the mRNA levels of the adhesion molecules *Icam1* and *Vcam1* in brain ECs isolated from *Trim47−/−* mice (Supplementary Fig. 14c). Interestingly, the Nrf2 activator diet prevented astrocyte activation in *Trim47*-deficient mice, as shown by GFAP staining in the hippocampus (particularly in the dentate gyrus) (Fig. 7d).

Finally, we assessed whether increasing the Nrf2 pathway could also restore the cognitive function of *Trim47−/−* adult mice. Behavioral tests revealed that reactivation of the Nrf2 pathway *via* a tBHQ diet was sufficient to normalize the performances of the *Trim47−/−* males in both the Y-maze (Fig. 7e, f) and the water maze tests (Fig. 7g, h and Supplementary Fig. 14d, e), indicating a reversal of cognitive deficits.

Together, these data demonstrate that enhancing the Nrf2 pathway was sufficient to prevent BBB leakage, brain defects and cognitive impairment in −/− mice. This highlights the important role of *Trim47* in maintaining brain physiology and cognitive function, primarily through the activation of the Nrf2 antioxidant pathway.

## Protein levels of NRF2 target genes are associated with MRI markers of cSVD

Finally, we measured and analyzed the protein level of NRF2 target genes in human plasma samples. In order to estimate the association of protein levels of the TRIM47/NRF2 pathway, we tested their association with MRI markers of cSVD in the 3C-Dijon and UK Biobank cohorts, focusing on white matter hyperintensities (WMH) and perivascular spaces (PVS) in white matter (WM) and basal ganglia (BG). Five proteins of the TRIM47/NRF2 pathway were available in our dataset and tested against WMH and PVS (HMOX1, GCLM, TXNRD1, GSR, and PRDX6) (Fig. 8a and Supplementary Table 1). Of those, higher levels of HMOX1 (HO1), TXNRD1 and PRDX6 showed consistent association with larger PVS in both BG and WM ($P < 0.05$). The association of higher level of GSR with larger BG-PVS and TXNRD1 with larger WM-PVS were significant after multiple testing correction (P-FDR < 0.05). Regarding WMH, higher level of PDRX6 were associated with smaller WMH volume, whereas TXNDR1 level were associated with larger WMH volume, both nominally ($P < 0.05$) (Fig. 8a and Supplementary Table 1).

Together, these human proteomic data indicate that analysis of the expression of partners of the NRF2 pathway could serve as a biomarker to potentially evaluate the severity and/or progression of cSVD and suggest that TRIM47/NRF2 pathway modulation could be predictive of a higher susceptibility to developing SVD.

## Discussion

In this study, we report that the cSVD risk gene *TRIM47* activates the NRF2 pathway in vitro and in vivo. TRIM47 controls brain EC resilience to OS by promoting NRF2 protein stability. In vivo, both global deletion and loss of endothelial *Trim47* result in a spontaneous vascular cognitive impairment phenotype characterized by increased BBB permeability, increased astrocyte activation in the hippocampus, especially the dentate gyrus and cognitive deficits in spatial recognition memory and spatial discrimination. Strikingly, activation of the NRF2 pathway using tBHQ treatment prevents these defects, showing that the protective TRIM47/NRF2 axis is required to ensure brain homeostasis (graphical model presented in Fig. 8b).

Despite progress in understanding genetic determinants of cSVD, specific signaling pathways leading to its development are still poorly understood. Current primary prevention strategies focusing on managing blood pressure and reducing risk factors are insufficient for slowing down the progression of cSVD and preventing related stroke and VCID. There is an urgent need to develop mechanism-based therapies for cSVD, which could have a major impact at the population level for the prevention of stroke and dementia and for promoting healthier brain aging[1,9,10]. Deciphering the molecular mechanisms underlying cSVD etiology is essential for identifying novel targetable pathways. We leveraged robust genetic evidence for a causal involvement of *TRIM47* loss-of-function variants in cSVD pathogenesis, previously obtained through a combination of genomic studies in large population-based cohorts, integrated with genetically determined gene expression in vascular and brain tissues, and screening of human knock-outs in a large biobank[21]. Proof of concept in vitro experiments had further supported TRIM47 as a strong causal gene for cSVD[21].

In the present study, we aim to investigate how this candidate selected from human studies could contribute to cSVD pathophysiology using preclinical models. We found that adult mice (males and females) deleted for *Trim47* display spatial memory deficits associated with increased BBB permeability (cadaverin, dextran, fibrinogen leakages) and decreased expression of tight junction proteins (Claudin5, Occludin) in cortical and hippocampal regions, suggesting that this pathology is driven by blood vessel damage developing at a young age. In line with this concept, endothelial-specific deletion of *Trim47* was able to fully mimic this phenotype, showing that the loss of *Trim47* in brain ECs is primarily responsible for the development of the disease. Interestingly, and supporting our findings, the genetic mouse model of endothelial Nitric Oxide Synthase (eNOS) deficiency which has been reported as a model of cerebral chronic hypoperfusion also causes spontaneous mild cognitive impairment associated with BBB leakage[53–55].

Additionally, we describe increased GFAP-positive astrocytes in *Trim47*-deleted mice, specifically in the dentate gyrus and in cortical regions where blood vessel leakiness was detected. This astrogliosis in *Trim47*-deficient mice is presumably due to the localized extravasation of plasma proteins such as fibrinogen. Supporting this hypothesis, previous elegant work from other groups has demonstrated the deleterious role of fibrinogen deposits into the CNS and its numerous neuropathological effects including astrocyte and microglia activation, demyelination and neurodegeneration[56–58].

Notably, Trim47−/− mice did not display microglial activation or overt neuroinflammation at 7-10 months under baseline conditions. While TRIM47 has been reported to activate NF-κB in models of acute lung injury[35] and cerebral ischemia[36], our findings suggest a context-dependent dual role, promoting either anti-inflammatory NRF2 signaling or pro-

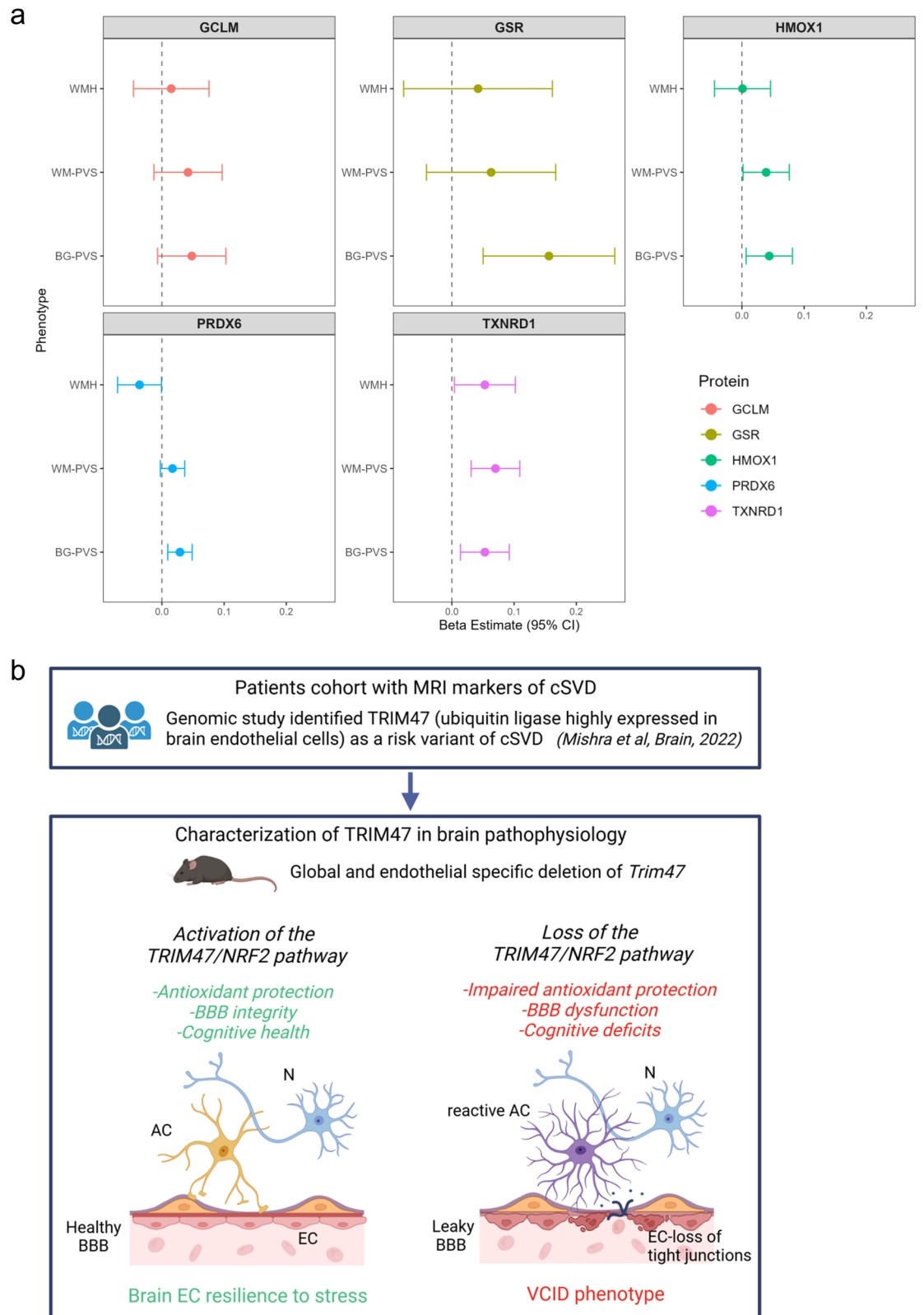

**Fig. 8 | Association of protein levels of TRIM47/NRF2 pathway with MRI markers of cSVD. a** Association of plasma protein levels with Magnetic Resonance Imaging (MRI) markers of cSVD: meta-analysis of 3C-Dijon and UK Biobank. *n* = 5,523. Forest plot of Bet estimates by protein and phenotype. WMH: white matter hyperintensities, PVS: perivascular spaces; BG: basal ganglia, WM: white matter. All graphical data are Beta estimate and 95% confidence interval (CI). **b** Graphical abstract. The TRIM47/NRF2 antioxidant pathway promotes BBB integrity, brain health and cognition. EC endothelial cells, AC astrocyte, N neuron, BBB blood–brain barrier, VCID vascular cognitive impairment and dementia. Created in BioRender (2026) https://BioRender.com/gbuzxad.

inflammatory NF-κB activation depending on physiological state and inflammatory severity (i.e., chronic/low *versus* acute/severe). Future transcriptomic studies will be needed to clarify TRIM47's role in regulating inflammation during later disease stages.

Our in vitro data identified TRIM47 as an important partner of the Nrf2 antioxidant system. TRIM47 acts upstream of NRF2 and activates this pathway by inducing NRF2 protein stability, potentially through binding to its inhibitor KEAP1 in brain ECs. Intriguingly, KEAP1 ubiquitination does not appear to be the primary mechanism through which TRIM47 limits NRF2 degradation, leaving the molecular basis of TRIM47-dependent KEAP1 regulation not elucidated. KEAP1 can also be inactivated through a non-ubiquitin mechanism mediated by the autophagy adaptor p62/SQSTM1 (reviewed in ref. 59). TRIM47 has been implicated in autophagy regulation[48], and consistent with this, our transcriptomic analysis of human brain ECs revealed reduced SQSTM1/p62 expression following TRIM47 depletion. These findings raise the possibility that TRIM47 promotes KEAP1 inactivation and subsequent NRF2 activation by enhancing SQSTM1/p62-dependent, autophagy-mediated KEAP1 turnover. This proposed mechanism warrants rigorous investigation in future cellular studies.

In vivo, our results confirmed TRIM47 regulation of the NRF2 system at postnatal age, in baseline and pathological conditions, but also in adult mice. Crucially, we demonstrated that activation of the Nrf2 pathway with the BBB-penetrant tBHQ is sufficient to prevent the BBB defects and the vascular dementia (VD) phenotype observed in *Trim47* mutant mice, proving that the endothelial TRIM47/NRF2 axis is a pivotal antioxidant mechanism actively promoting brain homeostasis. In fact, the brain vasculature is one of the main targets of localized OS processes. Reactive oxygen species and OS have been identified as factors with a potentially strong impact on the etiology of cerebrovascular diseases[60]. Although previously underestimated, age-associated vascular OS[61] and impairment of NRF2 signaling pathway[62] contributing to BBB disruption[63,64] are hypothesized to play a potential role in the pathogenesis of vascular dementia[65,66]. Nrf2 is a transcription factor that controls the expression of an array of antioxidant and detoxification enzymes[67] and has been shown recently to regulate the expression of tight junction proteins including Claudin5, even if the exact mechanism is not fully understood[45]. Importantly, Nrf2 deficiency was shown to exacerbate obesity-[68] or hypoperfusion-induced[69] BBB disruption and cognitive decline in mice. Conversely, Nrf2 activation with the antioxidant sulforaphane was demonstrated to improve BBB function after ischemic stroke[70]. These data support our in vivo findings showing that loss of *Trim47* induces Nrf2 pathway impairment, triggering primary endothelial damages, blood vessel insults, and leading ultimately to cognitive deficits.

Taken together, our data show that Trim47-deficient mice recapitulate key features of human cSVD, providing an innovative preclinical VCID model for investigating disease mechanisms. This model may also prove valuable for evaluating emerging therapeutic strategies targeting VD. As a limitation of this model, while *Trim47*−/− mice show blood vessel damages associated with cognitive deficits, no major WM lesion was detectable in young and middle-aged mice in hippocampus (5-12 months). The addition of other risk factors such as aging (examination of 24 month-old mice) or hypertension (Angiotensin II infusion model) could increase the susceptibility of the present model to develop measurable demyelination and axonal damages in hippocampus and other WM regions.

This study establishes a critical role for TRIM47 in brain physiology and implicates dysregulation of the TRIM47–NRF2 axis in the pathogenesis of cSVD. Our mouse and human data indicate that individuals carrying TRIM47 loss-of-function variants, genetically reduced TRIM47 expression, or altered NRF2 signaling may exhibit increased susceptibility to cSVD due to impaired antioxidant adaptive responses. These findings position NRF2 pathway activation as a promising therapeutic strategy for cSVD. Supporting this concept, treatment with tBHQ prevented the VD-like phenotype in Trim47-deficient mice. However, the known cytotoxic, genotoxic,

and carcinogenic liabilities associated with tBHQ (reviewed in ref. 71) preclude its clinical translation. Consequently, future work should focus on identifying and validating safer NRF2 activators or KEAP1 inhibitors, as well as alternative approaches targeting the TRIM47–NRF2 pathway, to enable the development of effective and clinically translatable interventions for cSVD and related VCID.

This work demonstrates the feasibility of using a combined bioinformatics, in vitro, and in vivo approach to functionally characterize robust genetic risk variants of cSVD. The present proof-of-concept study suggests that this strategy can efficiently inform drug discovery.

# Methods
## Study design
The objective of this study was to investigate the specific role of the cSVD genetic determinant TRIM47 in brain pathophysiology. We first identified the signaling pathways regulated by TRIM47 in endothelial cells (ECs) by performing RNA sequencing with validated siRNAs targeting TRIM47 on primary human brain microvascular endothelial cells (HBMEC). Bioinformatic analysis of these transcriptomic data revealed that the NRF2 antioxidant pathway was the most significantly repressed signaling pathway in TRIM47-deficient ECs. To investigate the molecular mechanisms by which TRIM47 promotes the NRF2 pathway, we employed a combination of complementary in vitro assays, including co-immunoprecipitation, promoter reporter assays in easily transfectable cell lines (HEK293, HeLa), as well as qPCR and Western blotting on HBMEC to assess the impact of TRIM47 loss or overexpression on NRF2, its inhibitor KEAP1, and NRF2 target genes. To further explore the effect of TRIM47 on OS, we performed in vitro experiments on HBMEC using CellRox, a fluorescent marker for ROS production. To investigate the role of TRIM47 in vivo, we used two mouse models: (1) *Trim47* knockout (KO) mice, in which *Trim47* is deleted in all tissues and organs to better model patient conditions, and (2) mice with endothelial-specific deletion of Trim47 following tamoxifen induction (*Trim47*[iEC-KO]) to investigate its specific role in ECs. To validate the antioxidant function of *Trim47* in vivo, we examined the brain and retina of *Trim47* KO mice after exposure to a well-established hypoxia-induced oxidative stress model (OIR) at postnatal day 16. We next assessed the impact of Trim47 loss on cognition using two well-characterized paradigms for spatial hippocampal memory (Y maze and Water maze). The potential contribution of anxiety and depression to the observed cognitive deficits was evaluated using the sucrose preference test. To elucidate the effect of *Trim47* deletion on blood vessels and other cell types (neurons, astrocytes, microglia, and inflammatory cells), we conducted histological analysis using immunofluorescence staining on brain sections from these mice, complemented by qPCR on brain extracts and isolated brain ECs. Functional assessment of the cerebral blood vessels was performed by evaluating blood–brain barrier permeability and detecting vessel leaks through complementary histological analysis. Additionally, we conducted transcriptomic analysis in mutant mice to investigate the modulation of the NRF2 pathway by TRIM47 in vivo. To test whether the observed phenotype in mutant mice was linked to the downregulation of the NRF2 signaling pathway, we treated mice with tBHQ. The route, dose, and duration of the treatment were chosen based on literature and preliminary data. This treatment approach was compatible with behavioral testing, as it minimized animal manipulation and associated stress. Finally, to assess the relevance of the NRF2 pathway in human disease, we performed proteomic analysis of plasma samples from patients with cSVD. This revealed a significant association between the protein levels of NRF2 target and MRI markers of cSVD. For all in vivo experiments, mice were randomly assigned to experimental groups using ALEA function in Excel. Study designs adhered to the principle of using the minimum number of animals sufficient to perform adequately powered statistical analysis, based on prior experience and animal ethics protocols. When possible, investigators were blinded to treatment and genotype. Sample size, type of replicates, and statistical methods used are detailed in the figures or figure legends. All in vivo and in vitro experiments were conducted in biological or technical replicates, as indicated for each experiment. No animals or

experimental units were excluded from the analyses and no a priori exclusion criteria were established.

## Cell culture and treatments

Human brain microvascular endothelial cells (HBMEC) were purchased from SicenCell (#1000) and cultured in Endothelial Cell Growth Medium-2 media (EGM-2) (Lonza). HBMECs were transfected with lentivirus or siRNA in EGM-2 media (Lonza). HeLa (ATCC CCL-2) and Hek293 (ATCC CRL-1573) were cultured in DMEM (Dulbecco's Modified Eagle Medium, Thermo Fisher Scientific) supplemented with 10% FBS (fetal bovine serum) and 1% penicillin-streptomycin (Thermo Fisher Scientific). Cells were transfected with a pool of two siRNA against TRIM47 (siTRIM47#1 and #2, denoted as siTRIM47 in the text) (15 nM for each TRIM47 siRNA, thus siRNAs at a final concentration of 30 nM) or with siRNA against NRF2 (15 nM) using INTERFERin® transfection reagent (Polyplus). Negative control siRNA (Eurogentec) was used, which is denoted as siCont. Sequences of the siRNA are listed in Supplementary Table 2. Plasmids were transfected using jetPRIME (Polyplus) following the manufacturer's protocol. Lentiviruses were transduced at a multiplicity of infection of 30.

For some experiments, cells were stimulated with H2O2 (Sigma), TBHP (Luperox®, Sigma) or Angiotensin II (ANGII, Sigma, A9525) to induce oxidative stress conditions. In some instances, HBMEC were treated with the proteasome inhibitor MG-132 for 4 h (5 μM, M7449, Sigma) to inhibit protein degradation. For rescue experiments, cells were transfected with siRNAs for 24 h before transduction with lentivirus encoding for human TRIM47. For activation of the NRF2 pathway, cells were pre-treated with tert-butylhydroquinone (tBHQ) (Sigma, 112941) or dimethyl fumarate (DMF) (Santa Cruz, sc-239774).

## Plasmids and lentiviruses

The plasmid pCMV6 TRIM47 (Myc-DDK-tagged) was purchased from Origene (RC218521). It contains the full sequence of human TRIM47 (NM_033452), which was then subcloned into the lentiviral vector pRRLsin-MND-MCS-WPRE. Preparations of lentivirus encoding for human TRIM47 were produced at the Bordeaux University lentivirus platform (TBMCore).

Flag-Keap1 was a gift from Qing Zhong (Addgene plasmid #28023; http://n2t.net/addgene:28023; RRID:Addgene_28023). pCDNA3-Myc3-Nrf2 was a gift from Yue Xiong (Addgene plasmid #21555; http://n2t.net/addgene:21555; RRID:Addgene_21555). pRK5-HA-Ubiquitin-WT was a gift from Ted Dawson (Addgene plasmid # 17608; http://n2t.net/addgene:17608; RRID:Addgene_17608).

pHOGL3/4.5 and pHOGL3/4.5 Triple mutant plasmids were a gift from Prof. Anupam Agarwal (University of Alabama at Birmingham, US). The pHOGL3/4.5 double mutant was obtained from the pHOGL3/4.5 triple mutant in which mutation C>T was created into the mutated EBox by a rapid-site-directed-mutagenesis using the following non-overlapping primers (F: cacgtgacccgccgagcata; R: ggccaggcggaacagc) and Platinum™SuperFi™II DNA Polymerase (Invitrogen). pGL3 Firefly Luciferase basic empty vector (Promega) was used as a control. pGL4.74 [hRluc/TK] vector] (Promega) Renilla luciferase vector was used as an internal control.

## Immunoblotting analysis and co-immunoprecipitation

HBMECs were lysed in RIPA buffer supplemented with protease and phosphatase inhibitors using a Tissue-Lyser (Qiagen). All protein lysates were then centrifuged for 15 min at $16000 \times g$ at 4 °C and supernatants were isolated. Proteins were loaded in 7 or 12% acrylamide gels and then transferred to a polyvinylidene difluoride (PVDF Immobilon membrane, Merck) membrane and incubated in PBS 0.1% Tween supplemented with 5% milk for 1 h to block non-specific binding. Immunoblots were next labeled with the primary antibodies listed in Supplementary Table 3. Primary antibodies were detected using fluorescently labeled secondary antibodies: anti-rabbit IgG Alexa Fluor™ 750 and anti-mouse IgG Alexa Fluor™ 700 (Fisher

Scientific). Detection of fluorescence intensity was performed using an Odyssey imaging system (Li-COR, ScienceTec) and Odyssey ver.3 software. Fluorescence intensity was quantified by densitometry using Fiji-ImageJ software to measure protein levels and normalized against loading controls. See Supplementary Figs. 15, 16 for the uncropped immunoblots.

For immunoprecipitation experiments, HEK293 cells were co-transfected with KEAP1-Flag (0.125 μg.ml−1) and TRIM47-myc-Flag (0.125 μg. ml−1). For semi endogenous and ubiquitin immunoprecipitation, HEK293 cells were co-transfected either with TRIM47-myc-Flag (0.45 μg. ml−1) or control empty vector (0.45 μg. ml−1) and HA-Ubiquitin (0.05 μg. ml−1). 48 hours post-transfection, cells were treated with DMSO (control), 10 μM MG132 (Sigma-Aldrich) and/or 0.1 μM MLN4924 (a selective NEDD8-activating enzyme inhibitor) (Sigma-Aldrich) for 4 h then lysed. Protein lysates were incubated with primary antibody at +4 °C, followed by incubation with Pierce™ Protein A/G Magnetic Agarose Beads (Thermo Fisher Scientific, 78609). Immunoprecipitation was carried out using primary antibodies at 2 μg for 500 μg of protein lysates. The primary antibodies used are listed in Supplementary Table 3.

## Reporter luciferase assay

HeLa cells were transfected with control or TRIM47 siRNA for 24 h or transfected with either 0.1 μg of TRIM47 plasmid or pcDNA3.1 empty vector and with HO1 luciferase reporter plasmids or pGl3 vector for 24 h. Luciferase activity was measured using the Dual-Luciferase® Reporter Assay System (Promega) and a Spark® Multimode Microplate Reader (Tecan). Luciferase reporter activity was normalized to the internal Renilla luciferase control and is expressed relative to the pGL3 empty vector.

## Real-time polymerase chain reaction

Total RNA from mouse tissues (brain and brain endothelial cells) and human cells (HBMEC and HeLa) was isolated by using the Direct-zol™ RNA MicroPrep kit (R2062, Zymo Research) and reverse transcribed into cDNA using the M-MLV Reverse Transcriptase (Promega). Quantitative real-time PCR was performed using the GoTaq qPCR master mix (Promega) on a QuantStudio™ 3 qPCR System (Thermo Fisher Scientific). Gene expression values were normalized to cyclophilin (PPIA) expression (human cells), *Pecam1* (mouse brain endothelial cells) or *18s* (mouse brain). See Supplementary Tables 4, 5 for the list of oligonucleotides.

## Bulk RNA-sequencing on HBMEC

HBMECs were transfected with control *vs* TRIM47 siRNA for 72 h. Total RNAs were isolated as described above. RNA-seq analysis was conducted on three biological replicates. Library preparation and sequencing were performed at the transcriptomic platform of the Paris Brain Institute (ICM). mRNA library preparation was performed based on the manufacturer's recommendations (KAPA mRNA HyperPrep Kit [ROCHE]). Pooled library preparations were sequenced on NextSeq® 500 whole genome sequencing (Illumina®), corresponding to 2×30 million reads per sample after demultiplexing. The quality of raw data was evaluated with FastQC. Poor-quality sequences were trimmed or removed with Trimmomatic software to retain only good quality paired reads. Star v2.5.3a 53 was used to align reads on the human GRCh37/hg19 reference genome using standard options. Quantification of gene was done using RNA-Seq by Expectation-Maximization (RSEM) v 1.2.28, prior to normalization using the edgeR Bioconductor software package. Data analysis and visualization of the results were done with the Data Analysis Core facility from Paris Brain Institute-ICM and their user interface (QuBy). Differential analysis was conducted with the generalized linear model framework likelihood ratio test from edgeR using a log2FC threshold set up to 1. Multiple hypothesis adjusted p-values were calculated using the Benjamini-Hochberg procedure to control for the false discovery rate (FDR). Genes with low counts (CPM < 5) were excluded from the analysis. Genes with an adjusted p-value (FDR) < 0.05 were considered as differentially expressed genes (DEG). Gene set enrichment analysis (GSEA) was performed using the Molecular Signatures Database (MSigDB) for the Wikipathways. Volcano plots and

dotplots with enrichment scores (NES) were directly downloaded from QuBy (http://quby.icm-institute.org).

## Oxidative stress measurement

HBMEC seeded on 24-well plates were transfected with control or TRIM47 siRNA and then replated on Nunc™ Lab-Tek™ II Chamber Slide™ (Thermo Fisher Scientific) after 24h. Cells were then pre-treated with DMSO, tBHQ (10 μM) or DMF (10 μM) for 24 h and then oxidative stress was induced by TBHP (Luperox, 150 μM, 1 h) or with angiotensin II (750 nM, 2 h). CellROX® Green reagent (C10444, Invitrogen) for oxidative stress detection and Hoechst for identifying nuclei (Thermo Fisher Scientific) were added directly to the live cells in complete EGM2 medium. After 30 min incubation, cells were washed 3 times in PBS and imaged with a fluorescent microscope (Observer Z.1, Zeiss). CellROX detects general ROS in live cells (including superoxide, hydroxyl radicals, and t-butyl hydroperoxide and serves as a general indicator of oxidative stress. Oxidative stress was measured by the quantification of green fluorescence in cytoplasm using the "Measure nuclear and cytoplasmic intensities tool" from ImageJ macro.

## Study approval

Animal experiments were approved by the local Animal Care and Use Committee of the Bordeaux University CEEA50 (IACUC protocol #36971). We have complied with all relevant ethical regulations for animal use. Mouse studies were conducted in accordance with the guidelines from Directive 2010/63/EU of the European Parliament on the protection of animals used for testing and research. All animals used were retained on a C57BL/6 background and both male and female mice were used for experiments. Animals received pre-emptive analgesia with buprenorphine (Vetergesic®) to minimize pain and distress. Adult mice were euthanized by administration of a lethal dose of injectable anesthetic (ketamine/xylazine), performed by authorized and trained personnel. For blood–brain barrier assessment procedures, an anesthetic eye drop (tetracaine 1%, single dose) was applied to the eye 5 minutes prior to injection, and fluorescent tracers were administered intravenously via retro-orbital injection under inhalation anesthesia (2% isoflurane). For brain histology procedures in adults, animals were placed under deep general anesthesia induced by ketamine/xylazine and euthanized by transcardial perfusion with PBS followed by PFA, resulting in exsanguination and confirmed death. For brain endothelial cell isolation, mice were euthanized by cervical dislocation performed under deep general anesthesia induced by ketamine/xylazine.

## Mice and breeding

Mice were housed in cages enriched with nesting and stimulation materials, including tubes, cotton, craft materials, and small wooden pieces. Animals were monitored daily for signs of stress, abnormal behavior, or weight loss. As an endpoint for the experiments, mice were humanely euthanized if they lost more than 20% of their initial body weight.

Trim47 full KO mice (C57BL/6N-Trim47em1(IMPC)Bay/Mmmh) were purchased from the national public repository system for mutant mice Mutant Mouse Resource and Research Center. The inducible endothelial-specific Trim47 knockout mouse model (Trim47iEC-KO) was generated by breeding Trim47fl/fl mice (strain ID: T005408, GemPharmtech) with Cdh5(PAC)-CreERT2 mice[72,73]. Endothelial deletion of Trim47 was induced in adult mice, by tamoxifen injection (5648, Sigma) (five injections of 0.5 mg daily). All experiments were conducted using littermate controls denoted in the text as Trim47 + /+ for the full KO mice and as Trim47iEC-WT for the endothelial-specific deleted mice line.

For the activation of the Nrf2 pathway, adult Trim47−/− and Trim47+/+ males were kept on a diet containing tBHQ (Sigma, 1% w/w). The food pellets were supplemented with tBHQ by the manufacturer, Safe. Body weight and food intake were monitored regularly. Behavioral tests were performed after a 1 month-diet and tissues were collected for histology, BBB permeability assessment or transcriptomic analysis after 2 months.

## Postnatal angiogenesis in mouse retina and oxygen-induced retinopathy model

Whole eyes at P6 and P12 were fixed in 4% PFA for 30 min at room temperature. Retinas were then dissected and blocked for 2 h in PBS containing 1% BSA and 0.3% Triton X-100. Retinas were washed with Pblec buffer (1 mM CaCl₂, 1 mM MgCl₂, 1 mM MnCl₂, 1% Triton X-100 in PBS), followed by overnight incubation at 4 °C with IB4-FITC and rabbit anti-ERG antibody in PBS supplemented with 3% fetal bovine serum (FBS), 1% donkey serum (DS) and 0.5% Triton. After washing, tissues were incubated with secondary antibody (donkey anti-rabbit Alexa Fluor 647) for 2 h at room temperature and then flat-mounted on glass slides using Fluoromount-G. Retinal vasculatures were imaged using a fluorescence microscope (Observer Z.1, Zeiss). The percentage of vascularization normalized to the whole retina area, the number of tip cells, and the number of veins and arteries were quantified using Zen Blue software (version 3.5, Zeiss).

The oxygen-induced retinopathy model (OIR) was carried out as described[74,75]. Mice pups and their nursing mother were exposed to 75% oxygen for 5 days from age P8 to age P13 in a hyperoxic chamber (ProOx Model 110, BioSpherix). At P13, they were returned to normal room air. Pups were euthanized at P16 by administration of a lethal dose of injectable anesthetic (ketamine/xylazine), to assess the degree of vascular regression (avascular area) and to evaluate retinal revascularization with pathological vessels (NV tufts). Only the pups weighing >4.5 g at P16 were included in the study since low weight impact the outcome and severity of the OIR model[76]. Mouse eyes at the P16 postnatal stage were dissected and stained with IB4-FITC in PBS supplemented with 3% FBS, 1% DS, and 0.5% Triton. Retinal vasculatures were imaged with a fluorescent microscope (Observer Z.1, Zeiss). Avascular retinal areas were quantified with Zen blue software, normalized to the total retinal area and expressed as a percentage compared to littermate controls.

## Behavioral tests

The Y-maze apparatus consisted of three arms (8 × 30 × 15 cm) separated by an angle of 120°. The recognition memory procedure took advantage of the innate tendency of rodents to explore novel environments. It consisted of four trials performed over 2 consecutive days. During the exploration (encoding) trial, mice were allowed to explore only two arms (starting arm and familiar arm) of the maze for 5 minutes, with the third arm being blocked (novel arm). The mice were given two exploration trials on the first day, with an inter-trial interval of 3h, and one additional trial on the second day to reinforce encoding. Three hours after the last exploration trial, mice underwent a recognition trial. Each mouse was put back in the starting arm of the maze, with free access to all three arms for 5 min. Time spent by mice in each arm and their trajectories were recorded and analyzed using an automated videotracking system (EthoVisionXT 16, Noldus). Discriminating the novel arm from the two familiar arms was used as an index of spatial recognition memory. Memory performance was expressed as the percentage of time spent in novel arm calculated as follows: (time spent in novel arm/time spent in all three arms) × 100.

The Morris water maze used a circular pool (180 cm in diameter) located in a dimly lit room decorated with various distal cues. The pool contained opaque water (22 ± 1 °C) and was subdivided into four virtual quadrants (A, B, C and D). A circular escape platform (10 cm in diameter) was submerged 1 cm beneath the water surface. The hidden platform remained in the same fixed position (quadrant D) during training. For each training trial, mice could swim for a maximum of 60 s to locate the hidden platform and were allowed to stay on it for 15 s upon finding it. Mice unable to locate the platform were gently guided to it. Mice were trained over 4 consecutive days and given 2 daily trials separated by an inter-trial interval of 3 h. The platform was then removed from the pool and probe trials were performed 3 days after the last training. The performance of each mouse was acquired and analyzed using an automated videotracking system using (EthoVisionXT 16, Noldus) to collect the following parameters: trajectory with heatmap, velocity, latency to the platform, distance to reach platform,

proximity measure, time spent in platform area, frequency on platform). *Trim47*. Search strategy use in water maze was analyzed manually as described in literature[44] by reviewing individual mouse trajectory for each trial during the training (days 1 and 4) and classified them into the three following categories: spatial/direct, scanning/random and looping/circling.

To assess anhedonia and anxiety, we performed a sucrose preference test on adult mice. The mice were previously accustomed to having 2 bottles in their cage, one week before the test. Mice were then offered two 50-ml bottles filled with water and 1% sucrose for 4 days. The 1% sucrose solution was then replaced by a 3% sucrose solution which was again replaced 4 days later by a 9% sucrose solution. Each day, water and sucrose amounts were monitored by weighing the water bottles, and the positions of the bottles in the cages were counterbalanced to avoid spatial preference. Sucrose preference was determined as the percentage of sucrose intake normalized to the total liquid intake.

### Endothelial cells isolation

For brain endothelial cells isolation, mouse brains collected from 2 adult mice were pooled and transferred into a petri dish containing cold HBSS without calcium/magnesium. Cerebellum and brain stem were then removed, leaving cerebral hemispheres intact. Each cerebral hemisphere was then cut in 4 sagittal slides before being enzymatically and mechanically digested using a gentleMACS™ Dissociator (#130-093-235, Miltenyi Biotec, 37C_ABDK_01 program) and the Adult Brain Dissociation kit (#130-107-677, Miltenyi Biotec) following the manufacturer. Brain lysates were filtered through a 70 μm MACS Smart Strainer (#130-198-462, Miltenyi Biotec). Incubation of the filtered suspension with Myelin Removal Beads II for 15 minutes (#130-096-733, Miltenyi Biotec) allowed depletion of myelin. ECs enrichment was performed by incubating the suspension with CD31 Microbeads for 30 minutes (#130-097-418, Miltenyi Biotec). The positive selection of ECs was done using LS column (#130-042-201, Miltenyi Biotec) placed on a magnetic separator (MACS® MultiStand and QuadroMACS™ Separator, Miltenyi Biotec). Endothelial cells were centrifuged (10 minutes, 500 g) and cell pellet was resuspended in 300 μl of TRI Reagent® (MRC gene) and then processed for RNA isolation.

For lung endothelial cells isolation, mouse lungs were collected from adult mice, minced with a scalpel and then digested using a gentleMACS™ (37C_m_LDK_1 program) and the Lung Dissociation kit (P) (Miltenyi Biotec, Cat#130-095-927). Lung lysates were filtered through a 70 μm MACS Smart Strainer. Endothelial cell enrichment was performed by incubating the suspension with CD31 Microbeads for 15 minutes (#130-097-418, Miltenyi Biotec). Endothelial cells were centrifuged (10 minutes, 500 g) and cell pellet was resuspended in TRI Reagent® and then processed for RNA isolation.

### Blood–brain barrier permeability assessment in vivo

To assess BBB permeability, fluorescent tracers were injected intravenously (retro-orbital) in adult (5-6 months) mice and left to circulate for 20 min as described in literature[77]. Cadaverine (0.95 kDa) conjugated to Alexa Fluor-555 (A30677, Invitrogen) was injected at a concentration of 100 μg Cadaverine/20 g of mice and lysine-fixable 3 kDa dextran conjugated to tetra-methylrhodamine (D3308, Invitrogen) at a concentration of 250 μg dextran/20 g. To assess tracers leak, a 300 μL blood sample was collected by cardiac puncture and centrifuged at 10 000 g, 10 min at 4 °C for serum preparation. Animals were then perfused in the left ventricle with warmed PBS. One hemi-brain (free of olfactory lobes and cerebellum) and the kidney were then collected, and their weight was measured before being snap-frozen in liquid nitrogen. Tissues were homogenized in PBS using a Tissue Lyser (Qiagen) and centrifuged at 15,000 g, 20 min, and 4 °C. Fluorescence measurement (RFUs) of 50 μL of diluted serum, brain and kidney supernatants was done using a Spark® Multimode Microplate Reader (Tecan). A Permeability Index (PI) was calculated as follows: PI (mL/g) = (Tissue RFUs/g tissue weight)/(Serum RFUs/mL serum) for each animal. The PI of each animal was divided with the mean PI of the *Trim47* +/+ group and data were represented as a ratio of PI. For additional histological analysis, the contralateral hemi-brain was directly OCT-embedded and preserved for immunohistochemistry.

### Immunofluorescence analysis of brain tissues

10 μm cryosections (sagittal) were prepared using a cryostat, then fixed with 4% PFA for 10 min at room temperature, blocked with blocking solution (PBS1X, 1% BSA, 0.5% Triton) for 1 h and incubated for 1.5 h at room temperature with primary antibodies diluted in buffer containing PBS1X, 0.5% BSA and 0.25% Triton X. For other histological analysis, brains were collected and placed in 4% PFA overnight at 4 °C. Brains were then washed 2 times 1 h with PBS1X and embedded in 4% agarose. 50 μm coronal sections were prepared using a Leica VT 1000S vibratome and placed in storage solution (Glycerol 30%, Ethylene glycol 30%, PBS-1×40%) −20 °C. Brain coronal sections were blocked with PBS 1X, 10% donkey serum, 0.05% Triton for 4 h prior to incubation with primary antibodies for 48 h and then with secondary antibodies (1/400 dilution). After 3×30 min wash with PBS1X, sections were mounted using mounting medium (Fluoromount G with DAPI, #00-4958-02, Thermo Fisher Scientific). Sections were imaged with a fluorescent microscope (Observer Z.1, Zeiss) and with a confocal microscope (LSM 900, Zeiss). Images were analyzed with Zen blue (version 3.5, Zeiss) and quantified with ImageJ (version 2.3.0, NIH).

### Human proteomic analysis

Association of plasma proteins of the NRF2/TRIM47 pathway with MRI-markers of cSVD was tested in the 3C-Dijon and UK Biobank cohorts. Informed consent of participants was obtained. All ethical regulations relevant to human research participants were followed. Study protocols were approved by the appropriate boards at their respective institutions: ethics committee of the University Hospital of Kremlin-Bicêtre for 3C-Dijon, and the National Research Ethics Service Committee North West-Haydock (reference 11/NW/0382) for UK Biobank.

The 3C-Dijon study is a population-based cohort study comprising 4,931 participants aged 65 years and older at inclusion recruited between 1999 and 2001[78,79]. A subset of 1,100 participants aged <80 years had both brain imaging (1.5 T Siemens Magneton scanner) and Olink® proteomic profiling, based on blood samples obtained at inclusion. Protein measurements were conducted on the Olink Explore 3072 panel using Proximity Extension Assay (PEA) technology, following the manufacturer's protocol[80], at McGill Genome Center (Montreal, Canada) capturing 2923 unique proteins across 8 protein panels[81]. Data pre-processing, including plate-based normalization and QC checks, was conducted according to standardized Olink protocols. WMH volume was estimated using a multimodal (T1, T2, DP) image processing algorithm[79]. PVS burden in basal ganglia and white matter was estimated with the previously described machine-learning-based SHIVA-PVS algorithm[17,82] using T1-weighted images. Proteomic profiling using Olink Explore 3072 and MRI measurements (3 T Siemens Prisma scanner) of the UK Biobank, a population-based cohort, were available for 5,523 participants. Analyses were conducted using WMH measurements (field ID: 25008) and PVS burden (in basal ganglia and white matter), estimated with the previously described machine-learning-based SHIVA-PVS algorithm[17,82] using T1-weighted images.

Five proteins of the NRF2/TRIM47 pathway were available in our dataset: HMOX1 (HO1), GCLM, TXNRD1, GSR and PRDX6. We conducted multivariable linear and logistic regression of individual proteins with WMH volume and PVS burden adjusted for the delay between age at blood draw and age at the time of MRI, sex, batch effect, total intracranial volume (or mask volume for WMH in 3C-Dijon). WMH volume and PVS burden in basal ganglia and white matter were inverse normal transformed. An inverse variance weighted meta-analysis was performed using *metafor* R package[83] to combine 3C-Dijon and UKB association analyses. The heterogeneity of associations across studies was assessed using the Cochran-Mantel-Haenszel statistical test; only associations with p > 0.01 (0.05/5) were considered. Benjamini-Hochberg correction for multiple testing was used; P-FDR < 0.05 was considered significant.

## Statistics and reproducibility

All in vitro results presented in this study are representative of at least three independent experiments. 'n' represents the number of biological replicates unless otherwise stated. All in vivo experiments were done on littermates with similar body weights per condition. Males and females were analyzed separately for behavioral tests. Data are shown as the mean ± standard deviation (s.d) as stated in the text. Statistical analysis was performed using GraphPad Prism 8 software. A two-sided unpaired t-test was performed for statistical analysis of two groups for in vitro data. A two-sided Mann–Whitney test was performed for the analysis of two groups for in vivo results. One-way ANOVA followed by Bonferroni's multiple comparisons test was performed for statistical analysis between 3 or more groups. Two-way ANOVA with multiple comparisons test was performed to evaluate mouse performances (2 or 3 groups) and progression over time during the Water maze. Chi square test was performed to compare the distribution of the search strategies between 2 groups during the water maze training. Differences were considered statistically significant with a P value < 0.05.

## Reporting summary

Further information on research design is available in the Nature Portfolio Reporting Summary linked to this article.

## Data availability

The raw reads for bulk RNA-sequencing in FASTQ format have been deposited in the Gene Expression Omnibus (GEO) database under accession number: GSE279052. The UK Biobank data used in this study were obtained from the UK Biobank under application no. 94113. Proteomic and MRI data are available upon reasonable request through the UK Biobank access procedures (www.ukbiobank.ac.uk). Due to confidentiality and ethical restrictions, individual-level proteomic data from the 3 C cohort are not publicly available but can be accessed upon reasonable request to the 3 C steering committee. Contact details for the data access committee are available via the study coordinating center (https://the-three-city-study-3c.com/index.php/confidentialite-et-droits/). All other data supporting the findings of this work are available within the paper and its Supplementary Information. Numerical source data underlying the graphs and charts from this paper can be found in Supplementary Data 1. Uncropped immunoblots are presented in Supplementary Fig. 15, 16. Any other data are available from the corresponding author upon reasonable request.

## Code availability

We used publicly available tools from OlinkAnalyze R package (v4.3.1, https://github.com/Olink-Proteomics/OlinkRPackage), ggplot2 R package (v3.5.2, https://github.com/tidyverse/ggplot2), forestplot R package (v3.1.7, https://cran.r-project.org/web/packages/forestplot) and SHIVA-PVS algorithm (https://github.com/pboutinaud/SHIVA_PVS, T1.PVS/v1).

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

## Acknowledgements

The authors thank Professor Anupam Agarwal (The University of Alabama at Birmingham, School of Medicine, United States) for providing the HO1 promoter luciferase constructs. We thank Philippe Alzieu, Sylvain Grolleau, Maxime David and Cédric Dupuy for their technical support and Christelle Boulle for administrative assistance. This work benefited from equipment and services from the Bordeaux Imaging Center (BIC) platform and the iGenSeq (RNA sequencing) and iCONICS (RNAseq analysis) core facilities at the ICM (Institut du Cerveau et de la Moelle épinière, PARIS, France). This project is supported by a grant overseen by the French National Research Agency (ANR) as part of the Investment for the Future Programme ANR-18-RHUS-0002 (SHIVA Project) and by the Precision and Global Vascular Brain Health Institute (VBHI) funded by the France 2030 IHU3 initiative.

## Author contributions

V.D., C.G. performed in vivo experiments and analyzed data; R.B., S.R, M.B., C.C and C.Proust performed in vivo experiments; JV and BJV contributed to experimental design and performed in vitro experiments and histology; E.C. performed histology and confocal imaging; J.L.M and B.B provided equipment for behavioral tests, analyzed and conceptualized data for behavior assessment; I.C. performed human data analysis, A.M. contributed to scientific discussion; S.D contributed to scientific discussion and manuscript writing; C.D. contributed to study design, conceptualized results and contributed to manuscript writing; T.C. designed the study, conceptualized results, and contributed to manuscript writing; C.Peghaire designed the study, performed experiments, analyzed and conceptualized results and wrote the manuscript.

## Competing interests

The authors declare no competing interests.
