## [Transparent Peer Review file · Communications Biology]

Endothelial TRIM47 regulates blood-brain barrier integrity and cognition via the KEAP1/NRF2 signalling pathway in mice

Corresponding Author: Dr Claire Peghaire

Version 0:

Reviewer comments:

Reviewer #1

(Remarks to the Author)

The manuscript by Dupl a et al entitled "Endothelial TRIM47 regulates blood-brain barrier integrity and cognition via the KEAP1/NRF2 signalling pathway" employed a multilayered experimental plan attempting to decipher the biological mechanisms underlying the association of TRIM47 with cSVD and provides overall compelling evidence that this was successful. Importantly, the current study is based on a strong foundation of bulk RNAseq data convincingly pointing towards the NRF2 pathway as Trim47 target. The experiments are carefully done, mostly stringently controlled and the manuscript is very well written and, overall, embedded in the state of the art in a well-balanced manner. In particular, the functional and in vivo data relating the Trim47/Nrf2 axis to cSVD and VCID are a strength of the manuscript. However, the uncovered mechanism still entails some key "loose ends" that should at least be discussed more comprehensively, ideally of course even addressed by additional experiments, of which one (endogenous or semi-endogenous CoIP of the Trim47/Keap complex) appears mandatory. To this end, while the authors have nicely deciphered the overall Trim47  Keap  Nrf2  antioxidant gene expression axis in ECs, the actual mechanism if and how Trim47 binds to and destabilizes Keap remains somewhat open.

Specific points:

1. Regarding the background/justification of the current study:

Was the initial genetic study that reported on Trim47 as a likely causal candidate, whose loss promotes cSVD severity (Brain 2022; "The most significant association with WMH volume and a composite extreme-cSVD phenotype was described at chr17q253. Within this gene-rich locus, summary-based Mendelian randomization and profiling of human loss-of-function allele carriers in UK Biobank pointed to TRIM47 as the most plausible causal gene with evidence for loss-of-function mechanisms driving the association with cSVD severity.") meanwhile confirmed by other (similar) genetic studies?

2. Trim47, like all Trims is a type I IFN-induced gene. Any idea/thoughts how this (sterile) inflammatory mechanism might play a role in brain ECs and BBB permeability? Along the same lines: what is the link between Trim47 and inflammation and could this play a role in cSVD? See also below.

How do the authors reconcile the recent identification of Trim47 as endothelial activation marker that promotes/aggravates inflammation via a K63-based signal transduction mechanism? (Qian, Y.; Wang, Z.; Lin, H.; Lei, T.; Zhou, Z.; Huang, W.; Wu, X.; Zuo, L.; Wu, J.; Liu, Y.; Wang, L.-F.; Guan, X.-H.; Deng, K.-Y.; Fu, M.; Xin, H.-B. TRIM47 Is a Novel Endothelial Activation Factor That Aggravates Lipopolysaccharide-Induced Acute Lung Injury in Mice via K63-Linked Ubiquitination of TRAF2. Signal Transduct. Target. Ther. 2022, 7 (1), 148. <https://doi.org/10.1038/s41392-022-00953-9>.)

The phrasing in the first Results text statement ("Our group, along with others, has recently reported that TRIM47 is not only expressed by various types of ECs, but is also particularly enriched in brain vessels and plays a crucial role in regulating ECs functions in vitro.") distorts somewhat the statements in Introduction (lines 76-81) on the same publications. I suggest to rephrase to better account for the contents of those publications.

As mentioned above, the RNAseq data hinting at the NRF2 pathway upon Trim47 deletion are compelling. But what about

the suggested terms related to cholesterol and SREB? Any suggestion how this might play into Trim47 functionalities in brain ECs?

The evidence that Trim47 binds to Keap is currently based on CoIP data after overexpression only. Authors should perform a CoIP from endogenous proteins, a semi-endogenous CoIP at the least.

Keap is the substrate receptor of Cul3Keap CRL-type E3 ligases. In Figure 2h, authors indicate an ubiquitination of Keap ("the substrate receptor within an E3 ligase") itself. Do authors suggest that binding of Trim47 renders Keap susceptible to ubiquitination by another E3 ligase? If so, which one? Is it Trim47 the E3 ligase? If so, the ubiquitination would be a K48-linked one? Is that the case? Or do authors imply that Keap is "auto-ubiquitinated" within the Cul3Keap CRL-type E3 ligase complex, rendering the complex inactive. Both is possible but would be associated with different outcomes. It is unclear to me which mechanism would be at play in brain ECs? One way to find out could e.g. be to employ MLN4924 (to block Cul3Keap CRL-type E3 ligase neddylation and thus activity).

Which redox species does CellRox detect specifically?

Re: "activation of the NRF2 pathway by stabilizing NRF2 protein with the tBHQ compound was able to dampen the oxidative stress induced by TRIM47 depletion under both basal and stress conditions": has it been shown that the E3 ligase activity of Cul3Keap CRL-type E3 ligases towards NRF2 is "counter-acted" by tBHQ? Please provide references! If there is no previous evidence that experiments would need to be done in the current revision.

The data on the lack of neuroinflammation in Trim47-KO mouse brains are somewhat weak. The inflammation screen is very limited – only 2x cytokines. Was bulk RNAseq done? Eventually scRNAseq would be needed to clarify involvement or non-involvement of inflammatory pathways in astrogliosis and BBB damage. This should at least be discussed and, without further evidence, the claim attenuated.

The manuscript contains some typos – please correct:

- Line 31-32: we propose that modulation of the TRIM47/NRF2 pathway could predict of a increased susceptibility to cSVD, suggesting
- Line 54: Genomics is powerful tool to identify
- Lines 185-186: Notably, the deletion of Trim47 resulted in a decreased in the Nrf2 pathway activation under both...
- Line 241: revealed a significantly increase in BBB leakiness
- And several others....

Reviewer #2

(Remarks to the Author)

This manuscript outlines a central molecular pathway through which TRIM47 regulates and maintains the health of brain endothelial cells; its loss or dysfunction could be detrimental to blood–brain barrier integrity. Overall, this is a thorough and solid piece of work describing the mechanism and potential therapeutic directions for cerebral small vessel disease. I do not have major criticisms, as the figures and data are presented in a clear and comprehensible manner. I agree with the authors' notion that the TRIM47-KEAP1-NRF2 pathway underlies the blood–brain barrier disruption phenotype observed in TRIM47-compromised conditions, including cerebral small vessel disease.

My minor comments are as follows:

- (1) The MWM probe test results for endothelial Trim47 KO mice (around Fig. 6) could be described even if not greatly significant; in that case, a sentence or even just p-values would suffice. [I leave this truly optional to the authors, as the manuscript is already thorough.]
- (2) Line 32: consider predict of a increased susceptibility to cSVD / predict the susceptibility to cSVD
- (3) Line 54: Genomics is "a" powerful tool to identify disease pathways and was/"has been" repeatedly shown to provide ...
- (4) BBB permeability assay: (Fig 5ab), Line 731-732. Is the histology reported in this paper? (This doesn't have to be addressed in the revision; I am just curious in case I missed it).
- (5) Fig 5e,c; which part of the hippocampus is this? DG (where GFAP upregulation is observed in Fig 5g?)

Reviewer #3

(Remarks to the Author)

Duplaa et al report on the involvement of the E3 ubiquitin ligase Trim47 in cerebral small vessel disease (cSVD). They previously identified TRIM47 as a cSVD risk gene in humans and herein report on Trim47 function in human endothelial cells in vitro and in mouse models of global and endothelial-specific Trim47 knockout mice. The data support functional involvement of Trim47 in brain homeostasis via protection of the blood brain barrier and oxidative stress regulation and reveal cSVD pathophysiology features in the knockout animals that can be rescued by the Nrf2 activator tBHQ. The work is comprehensive and well done and the data are supported by multiple lines of converging evidence. I have only a few suggestions for the authors to further improve their manuscript.

Specific comments:

Fig.1 and Fig.3 would benefit from a rescue experiment. If an si-RNA resistant Trim47 is expressed in Trim47 knockdown cells, is HO1 expression rescued? Is oxidative stress as measured by CellRox rescued?
Fig3 c-e: I don't think these experiments fully support the author's claim that Trim47 protects from oxidative stress. The observed changes in avascular area in the OIR experiment between controls and kos could be due to deficient

angiogenesis, changes in the immune system, etc, so this experiment does not directly test ROS signaling. The authors also measure gene expression changes in the brains of these animals. Immunostaining of identified targets such as HMOX1, GCLM could be informative here to test if the in vitro gene expression changes translate to mice. Can the authors isolate endothelial cells from these animals and run the CellRox assay in vitro? It would also be informative to see cells isolated from the EC-specific Trim47ko, as we expect endothelial Trim47 to be the anti-oxidative component. Minor: There's a typo in Fig3b panel (oxidative).

Fig.4 legend: the sentence 'Data revealed recognition memory in Trim47^{-/-} mice compared to Trim47^{+/+} mice' is incomplete and should be removed. Other sentences in the legend that contain interpretation of the data should also be removed. (d) pathes should be paths

Data in d-f are unclear to me: the heatmap is not very informative. E shows male mutant mice swim a longer distance to reach the platform, and f shows they swim at the same speed as controls. I would assume they take a longer time to reach the platform, but why is this datapoint not shown? Lastly can the authors comment on what could cause the sex specificity of the Trim47 response?

Fig5a,b please show actual values of cadaverine and dextran in the brains of the controls and knockouts.

Fig5d please include genes expressed in pericytes as a control for EC enrichment in the cell isolation procedure.

Fig5 c,e,g, h please add low magnification views of brain sections to help the reader understand if leakage is local and restricted to some brain areas or general throughout the brain.

Ref 69 should be updated.

Line 482 sentence unclear

Please comment on side-effects of tBHQ treatment.

Version 1:

Reviewer comments:

Reviewer #1

(Remarks to the Author)

This reviewer's concerns have been comprehensively and sufficiently addressed in the rebuttal letter and revised manuscript.

Reviewer #2

(Remarks to the Author)

The revision has addressed all of my previous comments, and I have no further suggestions or concerns. I support the manuscript in its revised form and appreciate the substantial effort the authors have invested in this work.

Reviewer #3

(Remarks to the Author)

The authors have addressed my previous comments and I am pleased to recommend publication of this article.

We thank all reviewers for their careful reading of our manuscript and for the constructive and insightful comments. We have addressed each point in detail and have revised the manuscript accordingly. Changes to the manuscript are highlighted in red in the revised version.

Reviewers' comments:

Reviewer #1 (Remarks to the Author):

The manuscript by Dupl a et al entitled "Endothelial TRIM47 regulates blood-brain barrier integrity and cognition via the KEAP1/NRF2 signalling pathway" employed a multilayered experimental plan attempting to decipher the biological mechanisms underlying the association of TRIM47 with cSVD and provides overall compelling evidence that this was successful. Importantly, the current study is based on a strong foundation of bulk RNAseq data convincingly pointing towards the NRF2 pathway as Trim47 target. The experiments are carefully done, mostly stringently controlled and the manuscript is very well written and, overall, embedded in the state of the art in a well-balanced manner. In particular, the functional and in vivo data relating the Trim47/Nrf2 axis to cSVD and VCID are a strength of the manuscript.

We thank the reviewer for their positive and constructive assessment. We are pleased that the rigor of our experimental design, the RNA-seq results, and the in vivo data linking endothelial TRIM47 to the KEAP1/NRF2 pathway in cSVD and VCID are recognized, as well as the clarity and relevance of the manuscript.

However, the uncovered mechanism still entails some key "loose ends" that should at least be discussed more comprehensively, ideally of course even addressed by additional experiments, of which one (endogenous or semi-endogenous CoIP of the Trim47/Keap complex) appears mandatory. To this end, while the authors have nicely deciphered the overall Trim47  Keap  Nrf2  antioxidant gene expression axis in ECs, the actual mechanism if and how Trim47 binds to and destabilizes Keap remains somewhat open.

Specific points:

1. Regarding the background/justification of the current study:

Was the initial genetic study that reported on Trim47 as a likely causal candidate, whose loss-of-function promotes cSVD severity (Brain 2022; "The most significant association with WMH volume and a composite extreme-cSVD phenotype was described at chr17q25. Within this gene-rich locus, summary-based Mendelian randomization and profiling of human loss-of-function allele carriers in UK Biobank pointed to TRIM47 as the most plausible causal gene with evidence for loss-of-function mechanisms driving the association with cSVD severity.") meanwhile confirmed by other (similar) genetic studies?

We thank the reviewer for the opportunity to clarify the background of our study. Early GWAS findings by Fornage, Debette et al. identified the chr17q25 locus as strongly associated with white matter hyperintensity (WMH) volume, suggesting a potential role in cerebral small vessel disease (cSVD), as WMH represents an established MRI marker of cSVD (Fornage et al., Annals of Neurology, 2011). Although TRIM47 was not initially identified as the causal gene, this study delineated the key genomic region involved. Subsequently, our consortium led by Prof. S. Debette refined these findings through a GWAS and whole-exome analysis of

an “extreme” cSVD phenotype (WMH + lacunes). By integrating summary-based Mendelian randomization and analyses of human loss-of-function allele carriers (UK Biobank data), we were the first to demonstrate that loss of TRIM47 expression is associated with increased cSVD severity, supporting TRIM47 as the most likely causal gene at the chr17q25 locus contributing to cSVD pathogenesis in humans (Mishra et al., Brain, 2022; reviewed in Bordes et al., Nature Reviews Neurology, 2022).

In the present work, we implemented a multilayered experimental approach combining in vitro and in vivo models to investigate the molecular mechanisms by which TRIM47 regulates endothelial homeostasis, blood–brain barrier function, and cognition, thereby contributing to cSVD development.

2. Trim47, like all Trims is a type I IFN-induced gene. Any idea/thoughts how this (sterile) inflammatory mechanism might play a role in brain ECs and BBB permeability?

Type I interferon (IFN) signaling exerts dual and context-dependent effects on the blood–brain barrier (BBB) and brain endothelial cells. Transient activation enhances antiviral defense and preserves barrier integrity, whereas sustained or excessive signaling promotes endothelial inflammation, tight junction disruption, and increased permeability. Consequently, the net outcome of IFN activity depends on both the intensity and duration of signaling, as well as the cellular context (Daniels & Klein, PLoS Pathog., 2015; Jana et al., Sci Rep., 2022; Zhang et al., J Neuroinflammation, 2023; Viengkhou & Hofer, Front Immunol., 2023).

As highlighted by the reviewer, several TRIM proteins (including TRIM5, TRIM6, TRIM14, TRIM21, TRIM22, and TRIM25) are transcriptionally induced by type I IFNs (reviewed in Chabot, Durantel, & Lucifora, PLoS Pathog., 2025). TRIM47, an E3 ubiquitin ligase, has been implicated in the regulation of innate immune responses across multiple contexts; however, its direct involvement in canonical type I IFN signaling in mammalian systems remains unsubstantiated. A recent study in hematopoietic stem cells reported that TRIM47 promotes K48-linked ubiquitination and degradation of MAVS, thereby attenuating downstream MAVS/IRF3/IKK–NF-κB signaling (Chen, Lu, Xu et al., Nat Commun., 2024).

In our Trim47 knockout mouse model, we have now quantified 3 canonical interferon-stimulated genes (Ifnar1, Ifnar2, and Isg15) by qPCR in brain endothelial cell and whole brain samples, noting that IFNs themselves were undetectable in these samples. The results (new **Figure 5d** and **Supplementary Fig 8e**) showed no evidence of pathway dysregulation in KO mice. Consistently, RNA-seq analysis of human brain microvascular endothelial cells (HBMECs) following TRIM47 depletion revealed no clear alteration in IFN signaling under baseline conditions (see heatmap below; provided for the reviewer only, not included in the revised manuscript).

RNA-sequencing data: heatmap showing no clear impact of TRIM47 depletion on interferon pathway in HBMEC, in baseline conditions.

Along the same lines: what is the link between Trim47 and inflammation and could this play role in cSVD? See also below. How do the authors re-concile the recent identification of Trim47 as endothelial activation marker that promotes/aggravates inflammation via a K63-based signal transduction mechanism? (Qian, Y.; Wang, Z.; Lin, H.; Lei, T.; Zhou, Z.; Huang, W.; Wu, X.; Zuo, L.; Wu, J.; Liu, Y.; Wang, L.-F.; Guan, X.-H.; Deng, K.-Y.; Fu, M.; Xin, H.-B. TRIM47 Is a Novel Endothelial Activation Factor That Aggravates Lipopolysaccharide-Induced Acute Lung Injury in Mice via K63-Linked Ubiquitination of TRAF2. *Signal Transduct. Target. Ther.* 2022, 7 (1), 148. <https://doi.org/10.1038/s41392-022-00953-9>.)

As pointed out by the reviewer, Qian et al. (reference 35 in the manuscript) reported a role for TRIM47 in NF- κ B-mediated inflammation in LPS-induced acute lung injury. They demonstrated that TRIM47 facilitates K63-linked ubiquitination of TRAF2 in endothelial cells, thereby activating NF- κ B and MAPK signaling, which promotes inflammatory cytokine production and monocyte adhesion in response to TNF α or LPS (Qian et al., *Signal Transduction & Targeted Therapy*, 2022).

In our study, we observed a pre-inflammatory state in 7-10-month-old *Trim47* knockout mice, characterized by reduced expression of endothelial tight junction proteins, increased BBB permeability, local astrogliosis, and cognitive deficits associated with decreased NRF2 pathway activity. At the timepoint examined, *Trim47*-deficient mice did not exhibit overt neuroinflammation and rather showed a trend toward decreased expression of *Icam1* and *Vcam1* in brain endothelial cells (**Figure 5d** and **Supplementary Fig 8e**), consistent with the pro-inflammatory role of TRIM47 suggested by Qian et al. Based on these findings, we believe that TRIM47 may exert a context-dependent dual role: it can either activate the protective, anti-inflammatory NRF2 pathway or promote the pro-inflammatory NF- κ B pathway, depending on the physiological context (baseline versus pathological conditions) and the magnitude or duration of inflammatory stimuli (chronic/low versus acute/severe).

Neuroinflammation is a key driver of brain diseases, highlighting the importance of clarifying the interplay between TRIM47, the IFN signaling pathway, and inflammation in brain endothelial cells and BBB function to better understand SVD pathophysiology. Further studies are needed to evaluate the contribution of IFN and inflammatory pathways to the *Trim47* knockout phenotype, particularly at later stages of disease progression (e.g., in aged mice) and/or under pathological conditions such as stroke or hypertension. While these questions are of considerable interest, addressing them would constitute an independent line of investigation and would warrant being the primary focus of a future study.

The phrasing in the first Results text statement ("Our group, along with others, has recently reported that TRIM47 is not only expressed by various types of ECs, but is also particularly enriched in brain vessels and plays a crucial role in regulating ECs functions in vitro.") distorts somewhat the statements in Introduction (lines 76-81) on the same publications. I suggest to rephrase to better accounts for the contents of those publications.

We apologize for the confusion caused by this sentence and for the incorrect references. The sentence has now been corrected and simplified to accurately reflect and align with the content of the subsequent paragraph (**page 3, lines 90–91**).

As mentioned above, the RNAseq data hinting at the NRF2 pathway upon Trim47 deletion are

compelling. But what about the suggested terms related to cholesterol and SREB? Any suggestion how this might play into Trim47 functionalities in brain ECs?

We thank the reviewer for this thoughtful comment and for highlighting the enrichment of cholesterol- and SREBP-related pathways in our RNA-seq data. Indeed, several differentially expressed genes upon Trim47 deletion are involved in cholesterol homeostasis and lipid metabolism and raft, processes tightly linked to endothelial cell function and BBB integrity (Zhu, Xiao et Wang, *Fluids Barriers CNS*, 2025). While our current study primarily focused on the NRF2-mediated oxidative stress response, we agree that the SREBP–cholesterol axis may represent an additional mechanism through which TRIM47 regulates endothelial homeostasis. Notably, SREBP signaling has been recently implicated in maintaining membrane integrity and tight junction composition, both essential for BBB stability (Tapia et al, *Dis Model Mech*, 2025). It is therefore plausible that TRIM47 influences BBB function partly by modulating lipid/cholesterol metabolic pathways and through crosstalk with oxidative stress in brain endothelial cells. This hypothesis could be investigated in future follow-up studies. Due to space limitations, we have not included this point in the Discussion of the revised manuscript.

The evidence that Trim47 binds to Keap is currently based on CoIP data after overexpression only. Authors should perform a CoIP from endogenous proteins, a semi-endogenous CoIP at the least.

We acknowledge this valid point raised by the reviewer. In response, we have now performed and incorporated new semi-endogenous CoIP experiments in the revised manuscript, in which TRIM47 was overexpressed in HEK293 cells and its binding to endogenous KEAP1 was examined. Our results, presented in **Supplementary Fig. 3a**, confirm the interaction between TRIM47 and endogenous KEAP1.

Keap is the substrate receptor of Cul3Keap CRL-type E3 ligases. In Figure 2h, authors indicate an ubiquitination of Keap (“the substrate receptor within an E3 ligase”) itself. Do authors suggest that binding of Trim47 renders Keap susceptible to ubiquitination by another E3 ligase? If so, which one? Is it Trim47 the E3 ligase? If so, the ubiquitination would be a K48-linked one? Is that the case?

We thank the reviewer for this insightful comment. Initially, our hypothesis was that TRIM47, as an E3 ubiquitin ligase, could directly induce ubiquitination of KEAP1. To test this, we performed ubiquitination assays in HEK293 cells (**Supplementary Fig. 3a**). Our data indicate that TRIM47 overexpression does not affect KEAP1 ubiquitination, and therefore, this is not the mechanism by which TRIM47 regulates KEAP1. Based on these results, we have revised our model and removed KEAP1 ubiquitination as a plausible TRIM47-dependent event (updated **Figure 2h**).

Importantly, previous studies have described an alternative mechanism of KEAP1 regulation that is p62/SQSTM1-dependent and involves autophagic inactivation of KEAP1. Interestingly, our RNA sequencing data in HBMECs show that TRIM47 regulates p62/SQSTM1 expression at the mRNA levels, suggesting that TRIM47 may influence KEAP1 through an autophagy-mediated mechanism. While this remains a hypothesis, we have updated the manuscript text in both the Results (**page 5**) and Discussion (**pages 13-14**) sections to reflect this revised mechanistic perspective.

Or do authors imply that Keap is “auto-ubiquitinated” within the Cul3Keap CRL-type E3 ligase complex, rendering the complex inactive. Both is possible but would be associated with different outcomes. It is unclear to me which mechanism would be at play in brain ECs? One way to find out could e.g. be to employ MLN4924 (to block Cul3Keap CRL-type E3 ligase neddylation and thus activity).

We thank the reviewer for this insightful suggestion and for proposing the use of MLN4924 to assess Cul3Keap CRL-type E3 ligase activity. We have now performed ubiquitination assays in HEK293 cells treated with MLN4924 (0.1 μ M, 4 h) and MG132 and included these data in **Supplementary Fig. 3b**. Our results do not provide evidence supporting a key role for neddylation, subsequent ubiquitination, or proteasomal degradation of Keap1 in the mechanism by which TRIM47 regulates Keap1 activity in brain endothelial cells. Accordingly, we do not believe that Cul3Keap CRL-type E3 ligase activity, or Keap1 “auto-ubiquitination,” significantly contributes in this context. These findings suggest that TRIM47 modulates KEAP1 through an alternative mechanism, possibly involving p62/SQSTM1-dependent, autophagy-mediated regulation as discussed above.

Which redox species does CellRox detect specifically?

To clarify, CellROX detects general reactive oxygen species (ROS) in live cells, including superoxide, hydroxyl radicals, and t-butyl hydroperoxide, but not reactive nitrogen species. Thus, it serves as a general indicator of oxidative stress rather than a probe for a specific redox species. These details have been now added to the Methods section-Oxidative stress Measurement, page 20.

Re: “activation of the NRF2 pathway by stabilizing NRF2 protein with the tBHQ compound was able to dampen the oxidative stress induced by TRIM47 depletion under both basal and stress conditions”: has it been shown that the E3 ligase activity of Cul3Keap CRL-type E3 ligases towards NRF2 is “counteracted” by tBHQ? Please provides references! If there is no previous evidence that experiments would need to be done in the current revision.

We thank the reviewer for this insightful comment. In fact, the ability of tBHQ to activate the NRF2 pathway by counteracting the E3 ligase activity of the Cul3–Keap1 complex toward NRF2 is well established in the literature. Mechanistically, tBHQ is an electrophilic compound that modifies specific cysteine residues on Keap1 (notably Cys151, and to a lesser extent Cys273, and Cys288), thereby disrupting the Keap1–NRF2 interaction and preventing Keap1-mediated ubiquitination and subsequent proteasomal degradation of NRF2. Previous studies collectively demonstrate that tBHQ inhibits the Cul3–Keap1 E3 ligase activity toward NRF2 by covalent modification of Keap1 cysteines (Takaya et al, *Free Radical Biology and Medicine*, 2012; reviewed in Suzuki, Takahashi, et Yamamoto, *Molecules and Cells*, 2023 (ref added to the revised manuscript)). Therefore, our use of tBHQ to activate the NRF2 pathway and rescue oxidative stress phenotypes upon TRIM47 depletion is based on established mechanisms and does not require additional validation experiments in this context.

The data on the lack of neuroinflammation in Trim47-KO mouse brains are somewhat weak. The inflammation screen is very limited – only 2x cytokines. Was bulk RNAseq done? Eventually scRNAseq would be needed to clarify to involvement or non-involvement of inflammatory pathways in astrogliosis and BBB damage. This should at least be discussed and, without further evidence, the claim attenuated.

We appreciate the reviewer's insightful comment regarding the neuroinflammatory status of *Trim47*-KO mouse brains. To assess the impact of *Trim47* deletion on inflammation, we initially performed immunostaining of brain sections for Iba1 and Cd68, as well as qPCR analysis of cerebral Cd11b expression, to evaluate microglial activation and immune cell infiltration (**Supplementary Fig. 12a-c**). Importantly, we have now expanded our analysis to include a broader panel of inflammation-related genes (*Icam1*, *Vcam1*, *Il1 β* , *Il6*, *Tnfa*, *Ifnar1*, *Ifnar2*, *Isg15*), examined by qPCR in brain endothelial cells isolated from adult *Trim47*^{+/+} and *Trim47*^{-/-} mice (**Figure 5d**) as well as in whole-brain lysates (**Supplementary Fig. 8e**). Although this screening is not exhaustive, it did not reveal a consistent pro-inflammatory signature in *Trim47*^{-/-} mice under baseline conditions.

We fully agree with the reviewer that single-cell RNA sequencing would provide a more comprehensive characterization of neuroinflammation and immune cell dynamics in these animals. While such experiments are of considerable interest, they are technically demanding and costly, and therefore beyond the scope of the present study. We consider them an important direction for future work. In line with the reviewer's suggestion, we have expanded the description of our results (**pages 9-10**) and moderated our discussion of *Trim47*'s role in inflammation (**page 13**).

The manuscript contains some typos – please correct:

- Line 31-32: we propose that modulation of the TRIM47/NRF2 pathway could predict of a increased susceptibility to cSVD, suggesting
- Line 54: Genomics is powerful tool to identify
- Lines 185-186: Notably, the deletion of *Trim47* resulted in a decreased in the *Nrf2* pathway activation under both...
- Line 241: revealed a significantly increase in BBB leakiness
- And several others....

We thank the reviewer for carefully pointing out these typographical errors. We have corrected all instances noted:

- Line 31–32: corrected to “predict an increased susceptibility”.
- Line 54: corrected to “Genomics is a powerful tool to identify”.
- Lines 185–186: corrected to “resulted in decreased *Nrf2* pathway activation”.
- Line 241: corrected to “revealed a significant increase in BBB leakiness”.

Additionally, we have carefully proofread the manuscript and corrected other typographical errors throughout to ensure clarity and consistency.

Reviewer #2 (Remarks to the Author):

This manuscript outlines a central molecular pathway through which TRIM47 regulates and maintains the health of brain endothelial cells; its loss or dysfunction could be detrimental to blood–brain barrier integrity. Overall, this is a thorough and solid piece of work describing the mechanism and potential therapeutic directions for cerebral small vessel disease. I do not have major criticisms, as the figures and data are presented in a clear and comprehensible manner. I agree with the authors' notion that the TRIM47-KEAP1-NRF2 pathway underlies the blood–brain barrier disruption phenotype observed in TRIM47-compromised conditions, including cerebral small vessel disease.

We thank the reviewer for the positive and encouraging evaluation of our manuscript. We appreciate the recognition of the TRIM47-KEAP1-NRF2 pathway as a central molecular mechanism in the regulation of brain endothelial cell function and blood–brain barrier integrity. The acknowledgment of the clarity and rigor of our data presentation is highly valued. This feedback reinforces the relevance of our findings to the pathophysiology of cerebral small vessel disease and supports their potential translational implications.

My minor comments are as follows:

(1) The MWM probe test results for endothelial Trim47 KO mice (around Fig. 6) could be described even if not greatly significant; in that case, a sentence or even just p-values would suffice. [I leave this truly optional to the authors, as the manuscript is already thorough.]

We thank the reviewer for this thoughtful suggestion. In line with the recommendation, we have now included the results of the MWM probe test for endothelial Trim47-KO mice in the revised manuscript, which show a trend toward an increased proximity measure (mean distance to platform) (**Figure 6h**). Although the differences observed were not statistically significant, we have reported the corresponding p-value ($p=0.0597$). The lack of statistical significance likely reflects variability in knockout efficiency inherent to the Cre-line approach, in contrast to the more consistent effects observed with constitutive, full Trim47 deletion. This addition ensures that the behavioral assessment is fully documented, even though the outcome does not indicate a strong effect.

(2) Line 32: consider predict of a increased susceptibility to cSVD / predict the susceptibility to cSVD

(3) Line 54: Genomics is "a" powerful tool to identify disease pathways and was/"has been" repeatedly shown to provide ...

We thank the reviewer for carefully pointing out these typographical errors. We have corrected all instances noted:

- Line 31–32: corrected to “predict an increased susceptibility”.
- Line 54: corrected to “Genomics is a powerful tool to identify”.

(4) BBB permeability assay: (Fig 5ab), Line 731-732. Is the histology reported in this paper? (This doesn't have to be addressed in the revision; I am just curious in case I missed it).

We thank the reviewer for this question. We attempted to assess BBB permeability using cadaverin and small dextran extravasation by histology; however, we were unable to obtain

reliable signals, most likely due to technical issues related to fixation and/or permeabilization, despite following a validated protocol (Devraj et al., J. Vis. Exp., 2018). To evaluate BBB permeability more robustly, we instead performed histological analysis of plasmatic fibrinogen extravasation, which yielded consistent results (Figure 5c). In addition, as suggested by another reviewer, we have included wide-field images in the revised manuscript to complement these data and to provide a broader visualization of the local vascular leakages observed in Trim47-KO mice across cortical regions, hippocampus, basal ganglia, and corpus callosum (**Supplementary Fig. 9a-b**).

(5) Fig 5e,c; which part of the hippocampus is this? DG (where GFAP upregulation is observed in Fig 5g?)

We appreciate the reviewer for raising this important point. Figures 5c and 5e highlight the increased fibrinogen extravasation together with the concomitant reduction in claudin-5 staining observed in hippocampal regions, particularly within the CA1 area. In addition, we have included wide-field images of fibrinogen leakage (**Supplementary Fig. 9a-b**) and claudin-5 staining (**Supplementary Fig. 10**), which further illustrate vascular defects across several cortical regions as well as in hippocampal areas (CA1 and DG). Notably, astrocyte hyperactivity in the hippocampus was detected only in the DG region in Trim47-KO mice, but not clearly in the CA1 area (Figure 5g), suggesting a region-specific vulnerability to astrogliosis following the loss of *Trim47*.

Reviewer #3 (Remarks to the Author):

Duplaa et al report on the involvement of the E3 ubiquitin ligase Trim47 in cerebral small vessel disease (cSVD). They previously identified TRIM47 as a cSVD risk gene in humans and herein report on Trim47 function in human endothelial cells in vitro and in mouse models of global and endothelial-specific Trim47 knockout mice. The data support functional involvement of Trim47 in brain homeostasis via protection of the blood brain barrier and oxidative stress regulation and reveal cSVD pathophysiology features in the knockout animals that can be rescued by the Nrf2 activator tBHQ. The work is comprehensive and well done and the data are supported by multiple lines of converging evidence. I have only a few suggestions for the authors to further improve their manuscript.

We thank the reviewer for the constructive and encouraging evaluation of our study. We appreciate the recognition of TRIM47 as a key regulator of brain endothelial cell function, blood–brain barrier integrity, and oxidative stress responses. We are pleased that the comprehensive nature of our work and the converging lines of evidence were noted, including the demonstration that Nrf2 activation by tBHQ can rescue the phenotype observed in Trim47-deficient conditions.

Specific comments:

Fig.1 and Fig.3 would benefit from a rescue experiment. If an si-RNA resistant Trim47 is expressed in Trim47 knockdown cells, is HO1 expression rescued? Is oxidative stress as measured by CellRox rescued?

We appreciate the reviewer's recommendation and have addressed it accordingly. In line with this suggestion, we performed rescue experiments by overexpressing a lentiviral TRIM47 construct in TRIM47-knockdown human brain microvascular endothelial cells. Re-expression of TRIM47 restored HO1 and NQO1 levels and significantly reduced oxidative stress, as measured by CellROX dye, confirming the specificity of the knockdown phenotype. These new data have been incorporated into the revised manuscript (**Supplementary Fig. 1d and Supplementary Fig. 4c–d**), further supporting the role of TRIM47 in regulating NRF2-target gene expression and oxidative stress responses.

Fig3 c-e: I don't think these experiments fully support the author's claim that Trim47 protects from oxidative stress. The observed changes in avascular area in the OIR experiment between controls and kos could be due to deficient angiogenesis, changes in the immune system, etc, so this experiment does not directly test ROS signaling. The authors also measure gene expression changes in the brains of these animals. Immunostaining of identified targets such as HMOX1, GCLM could be informative here to test if the in vitro gene expression changes translate to mice.

We appreciate the reviewer's insightful critique. To our knowledge, there is currently no pure or exclusive in vivo model of oxidative stress in mice, as oxidative stress is typically intertwined with other biological processes such as angiogenesis, immune regulation, and metabolism. We agree that the OIR experiment alone does not directly test ROS signaling and that alternative mechanism, including impaired angiogenesis or immune alterations, could contribute to the observed changes in avascular area. To exclude a potential contribution of angiogenesis to this phenotype, we have now included in the revised manuscript an analysis of postnatal angiogenesis in P6 and P12 retinas collected from Trim47-WT and Trim47-KO mice. Our data indicate that the loss of Trim47 is not associated

with delayed angiogenesis, impaired sprouting, or artery/vein specification, demonstrating that Trim47 is dispensable for postnatal angiogenesis in the mouse retina (**Supplementary Figure 6**). As pointed by the reviewer, we agree that we cannot exclude the potential role of immune cells and inflammation to the OIR phenotype and thus decided to tone down our conclusion in the results section (pages 6 and 7).

To strengthen our data, and as suggested by the reviewer, we have performed additional immunostaining for the key oxidative stress-related target HO1 in brain sections from Trim47-KO mice. Because HO1 displays broad expression, we focused our quantification on vascular HO1, specifically where HO1 colocalized with podocalyxin expression. Our data indicate that vascular HO1 protein levels are modestly (~30%) but significantly downregulated in Trim47-KO mice in cortical regions (**Supplementary Figure 11**). These experiments confirm that the gene expression changes observed in vitro are recapitulated in vivo, thereby supporting the role of Trim47 in protecting against oxidative stress.

Can the authors isolate endothelial cells from these animals and run the CellRox assay in vitro? It would also be informative to see cells isolated from the EC-specific Trim47ko, as we expect endothelial Trim47 to be the anti-oxidative component.

We acknowledge the reviewer's thoughtful suggestion. However, isolating primary endothelial cells from Trim47-KO and EC-specific Trim47-KO mice for subsequent CellROX assays is not feasible within the scope of this study due to the considerable time, cost, and animal resources required. Such experiments would require additional cohorts of mice and extensive cell isolation procedures, which go beyond the current project framework. Instead, we have focused on complementary in vivo and histological approaches that provide robust evidence for the role of Trim47 in oxidative stress regulation.

Minor: There's a typo in Fig3b panel (oxidative).

Fig.4 legend: the sentence 'Data revealed recognition memory in Trim47^{-/-} mice compared to Trim47^{+/+} mice' is incomplete and should be removed. Other sentences in the legend that contain interpretation of the data should also be removed. (d) paths should be paths

We thank the reviewer for carefully pointing out these typographical errors. We have corrected all instances noted.

Data in d-f are unclear to me: the heatmap is not very informative.

E shows male mutant mice swim a longer distance to reach the platform, and f shows they swim at the same speed as controls. I would assume they take a longer time to reach the platform, but why is this datapoint not shown? Lastly can the authors comment on what could cause the sex specificity of the Trim47 response?

We thank the reviewer for raising this point regarding the heatmaps presented in Figure 5. We believe that showing mouse paths from the first and last day of the training phase in the Morris Water Maze (MWM) task is informative. These heatmaps illustrate where the mice spent most of their time and capture learning progression: a healthy mouse typically shows random search patterns during early trials (day 1) and increasingly focused searches near the platform location during later trials (day 4). Heatmaps are also valuable because they can reveal motor- or anxiety-related confounds and group differences that quantitative metrics alone may not fully capture. Additionally, they help readers visualize representative search

strategies (e.g., looping, scanning, or spatial strategies) used by Trim47 WT and KO mice, as quantified in Supplementary Fig. 7g–h.

We apologize for any potential confusion regarding the MWM results. To clarify the data presented in Figure 5:

(c) shows the latency (**time**) to reach the hidden platform across the 4 training days and reveals a significant increase in Trim47 KO mice compared with Trim47 WT mice on day 4.

(d) presents representative heatmaps showing mouse paths during the first and last day of training.

(e) shows the **distance** traveled to reach the hidden platform on day 4 and confirms the latency findings, as Trim47 KO mice swam a longer distance to reach the platform.

(f) shows swim **speed** on day 4 and reveals no difference between Trim47 KO and WT mice, confirming that KO animals exhibit cognitive-but not locomotor-deficits in the MWM.

We analyzed males and females separately in the MWM (and Y-maze) because biological sex can influence exploratory behavior, anxiety levels, and learning strategies, all of which directly affect performance in these cognitive tasks. Assessing each sex independently improves the accuracy, interpretability, and reproducibility of the behavioral results, in line with current guidelines for rigorous experimental design. Importantly, we want to clarify that no sex-specific phenotype was observed in Trim47 KO mice, as explicitly stated in the Results section of the manuscript (page 7, lines 228–230).

Fig5a,b please show actual values of cadaverine and dextran in the brains of the controls and knockouts.

We acknowledge the reviewer's request and have now provided the raw fluorescence values and the non-normalized permeability index for cadaverine and dextran in the brains of Trim47 WT and KO mice in **Supplementary Fig. 8c and 8d**.

Fig5d please include genes expressed in pericytes as a control for EC enrichment in the cell isolation procedure.

We thank the reviewer for this suggestion. Although our isolation protocol is designed for ECs, it can also capture pericytes due to their close association with ECs. Consistent with this, **Supplementary Fig. 5g** shows mild enrichment of pericyte-specific genes (Pdgfrb and Anpep) in brain EC samples compared to whole-brain samples, strong enrichment of endothelial markers as expected (Cd31, Cdh5), and loss of neuronal and myelin markers (Tuj1, S100b), supporting the specificity of the EC population analyzed. Additionally, expression of pericyte-specific genes (Pdgfrb and Anpep) was included in **Supplementary Fig. 8e** to assess pericyte content in brains from Trim47 WT and KO mice.

Fig5 c,e,g, h please add low magnification views of brain sections to help the reader understand if leakage is local and restricted to some brain areas or general throughout the brain.

We thank the reviewer for this important question. We have now provided low-magnification images of the brain sections as requested. These images reveal subtle fibrinogen vascular leakage (**Supplementary Fig. 9a–b**) and focal loss of Claudin-5 staining (**Supplementary Fig. 10**) across various brain regions, including the hippocampus (DG and CA1), cortical

regions, basal ganglia, and white matter (corpus callosum). In contrast, astrocyte hyperactivity is largely confined to the DG of the hippocampus and cortical regions, which may be more vulnerable to vascular leakage. This demonstrates that the vascular alterations are not uniformly distributed but occur in discrete, region-specific areas, providing a clearer context for the high-magnification images shown in Fig. 5. If beyond the scope of the present study, a comprehensive analysis of potential white matter lesions in Trim47 KO mice-using a combined macro- and micro-scale approach with MRI/DTI and electron microscopy-will be necessary to fully elucidate the impact of Trim47 deletion on WM and will be addressed in future investigations.

Ref 69 should be updated.

A more recent publication using the using the Cdh5(PAC)-CreERT2 line generated by Ralf Adams has been added ((72) Sørensen et al; *Blood*, 2009, and new ref (73) Benz et al, *eLife* 2019).

Line 482 sentence unclear

We apologize for the confusion. The sentence has been corrected in the revised manuscript.

Please comment on side-effects of tBHQ treatment.

We acknowledge the importance of the point raised by the reviewer. Indeed, treatment with the BBB-penetrant compound tBHQ in our study provided proof of concept that pharmacological upregulation of the NRF2 pathway can enhance BBB integrity and improve cognitive performance in Trim47-deficient mice. However, tBHQ presents significant limitations that restrict its translational and clinical potential. Although widely used as a preservative and food additive, long-term or high-dose exposure to tBHQ has been associated with cytotoxic, genotoxic, and carcinogenic effects (reviewed in Khezerlou et al., *Toxicol. Rep.* 2022), making it unsuitable for chronic therapeutic use or clinical application. As now discussed in the revised manuscript (page 15), future studies validating safer NRF2 activators or KEAP1 inhibitors, as well as other pharmacological and genetic strategies targeting the TRIM47–NRF2 pathway, will be required to develop effective and clinically translatable interventions for cSVD and related VCID.

REVIEWERS' COMMENTS:

Reviewer #1 (Remarks to the Author):

This reviewer's concerns have been comprehensively and sufficiently addressed in the rebuttal letter and revised manuscript.

Reviewer #2 (Remarks to the Author):

The revision has addressed all of my previous comments, and I have no further suggestions or concerns. I support the manuscript in its revised form and appreciate the substantial effort the authors have invested in this work.

Reviewer #3 (Remarks to the Author):

The authors have addressed my previous comments and I am pleased to recommend publication of this article.

We would like to sincerely thank all Reviewers for their thorough evaluation of our manuscript and for their very positive feedback. We are grateful to Reviewer #1 for confirming that all concerns have been comprehensively and satisfactorily addressed in both the rebuttal letter and the revised manuscript. We also thank Reviewer #2 for acknowledging that all previous comments have been fully addressed and for their support of the revised version, as well as their appreciation of the substantial effort invested in this work. Finally, we are thankful to Reviewer #3 for their positive assessment and recommendation for publication.

We greatly appreciate the reviewers' time, careful consideration, and constructive input, which have significantly contributed to improving the quality, clarity, and overall strength of the manuscript.